# LAW OF VISION REPRESENTATION IN MLLMS

## ABSTRACT

We present the "Law of Vision Representation" in multimodal large language models (MLLMs). It reveals a strong correlation between the combination of cross-modal alignment, correspondence in vision representation, and MLLM performance. We quantify the two factors using the cross-modal **A**lignment and **C**orrespondence score (AC score). Through extensive experiments involving thirteen different vision representation settings and evaluations across eight benchmarks, we find that the AC score is linearly correlated to model performance. By leveraging this relationship, we are able to identify and train the optimal vision representation only, which does not require finetuning the language model every time, resulting in a 99.7% reduction in computational cost.

## 1 INTRODUCTION

Current multimodal large language models (MLLMs) (Chen et al., 2024a; Liu et al., 2024e;d) have achieved remarkable advancements by integrating pretrained vision encoders with powerful language models (Touvron et al., 2023; Zheng et al., 2023). As one of the core components of a general MLLM, the vision representation is critical. Many researchers have utilized CLIP (Radford et al., 2021) as the primary image feature encoder, but its limitations are increasingly noticeable (Tong et al., 2024b; Geng et al., 2023; Yao et al., 2021). As a result, alternative vision representations and the combination of multiple vision encoders are being actively explored (Tong et al., 2024a; Lin et al., 2023).

Despite this growing attention, the selection of vision representation has largely been empirical. Researchers typically test a set of vision representations on a specific MLLM and choose the one that yields the highest performance on benchmark tasks. This approach, however, is constrained by the number of representations tested and does not address the underlying factors that make certain feature representations perform better than others. As a result, the optimal vision representation for a specific MLLM is often determined by empirical performance rather than a deep understanding of the features that contribute to success. The question of what fundamentally makes a feature representation achieve the highest performance remains largely unanswered.

To address this gap in understanding what makes a vision representation optimal for MLLMs, we propose the **Law of Vision Representation in MLLMs**. It aims to explain the key factors of vision representation that impact MLLM benchmarks performance. Our findings reveal that *the cross-modal **Alignment** and **Correspondence** (AC) of the vision representation are strongly correlated with model performance.* Specifically, an increase in the AC of the selected vision representation leads to improved model performance. To quantify this relationship, we define an **AC score** that measures cross-modal alignment and correspondence in vision representation. The AC score and model performance exhibit a linear relationship, with a coefficient of determination of 95.72%.

Furthermore, the Law of Vision Representation guides the selection of an optimal vision representation for MLLMs. Originally, this process was extremely costly because even subtle changes in vision encoding methods—such as switching encoder types, altering image resolution, or testing feature combinations—require finetuning the language model (Lin et al., 2024). For example, using a top data-efficient MLLM pipeline with a 7B language model requires 3,840 NVIDIA A100 GPU hours to test the 10 encoders used in this study, amounting to a cost of approximately $20,000[1]. Testing additional encoders leads to a linear increase in cost. Moreover, the recent trend of feature combination, which often results in better performance, necessitates combinatorial testing of vision

---

[1]https://replicate.com/pricing

Figure 1: Visualization of the Law of Vision Representation in MLLMs.

encoders. Testing all possible combinations of 10 encoders results in 1023 combinations, exponentially increasing the cost and energy consumption. This process consumes approximately 100,000 kilowatt-hours[2], enough to drive an electric vehicle around the Earth 13 times.

Thus, we are the first to propose a policy, **AC policy**, that selects the optimal vision representation using AC scores within the desired search space. Unlike traditional methods that rely on benchmarking performance, the AC policy enables the expansion of the search space—allowing for an increased number of vision representations to be considered—without incurring additional costs. We demonstrate that this approach enhances both accuracy and efficiency compared to randomly searching for the optimal representation. The policy successfully identifies the optimal configuration among the top three choices in 96.6% of cases, with only three language model finetuning across a 13-setting search space.

## 2 RELATED WORKS

### 2.1 VISION FOR MLLMs

Recent studies have explored various vision representations in MLLMs (Beyer et al., 2024; Ge et al., 2024; Liu et al., 2024e; Wang et al., 2024b; Sun et al., 2023; Luo et al., 2024). Interestingly, some findings indicate that relying solely on encoders outside of the CLIP family (Cherti et al., 2023; Zhai et al., 2023b; Li et al., 2022), such as DINOv2 (Oquab et al., 2023) and Stable Diffusion (Rombach et al., 2021), often leads to lower performance scores (Karamcheti et al., 2024; Tong et al., 2024a). However, combining features from these encoders with CLIP features—such as concatenating image embeddings in the token or channel dimension—significantly enhances performance beyond using CLIP alone (Tong et al., 2024a;b; Liu et al., 2024c; Kar et al., 2024). Researchers intuitively suggest that these additional encoders provide superior detail-oriented capabilities, but no studies have thoroughly analyzed the underlying causes of the performance change (Wei et al., 2023; Lu et al., 2024a). This suggests that the attributes of an optimal vision representation remain not fully understood.

---

[2]https://www.nvidia.com/content/dam/en-zz/Solutions/Data-Center/a100/pdf/nvidia-a100-datasheet.pdf

## 2.2 CROSS-MODAL ALIGNMENT

Cross-modal alignment refers to the alignment between image and text feature spaces (Duan et al., 2022). This concept emerged with the introduction of text-image contrastive learning (Radford et al., 2021; Jia et al., 2021). Although current MLLMs utilize contrastively pretrained image encoders, the challenge of achieving effective alignment persists (Ye et al., 2024; Zhai et al., 2023a; Woo et al., 2024). Despite efforts to critique the limitations of CLIP family representations and explore alternative vision representations, many approaches continue to rely on contrastively pretrained encoders or adding contrastive loss without fully eliminating them (Zhang et al., 2024b; Lu et al., 2024a; Tong et al., 2024a;b; Liu et al., 2024b). In our work, we point out that alignment in vision representation is essential for improved model performance and is crucial for data efficiency. Without pre-aligned vision representations, extensive data pretraining is required to achieve cross-modal alignment within the language model (Ge et al., 2024; Chen et al., 2024b; Li et al., 2024c).

## 2.3 VISUAL CORRESPONDENCE

Visual correspondence is a fundamental component in computer vision, where accurate correspondences can lead to significant performance improvements in tasks, such as image detection (Xu et al., 2024; Nguyen & Meunier, 2019), visual creation (Tang et al., 2023; Zhang et al., 2024c), and MLLMs (Liu et al., 2024a), etc. Correspondences are typically categorized into semantic- and geometric-correspondences. Semantic correspondences (Zhang et al., 2024c; Min et al., 2019) involve matching points that represent the same semantic concept not necessarily representing the same instance. Geometric correspondences (Sarlin et al., 2020; Lindenberger et al., 2023), on the other hand, require matching the exact same point across images, which is often crucial for low-level vision tasks, such as pose estimation (Sarlin et al., 2020; Lindenberger et al., 2023; Zhang & Vela, 2015),and SLAM tasks, etc.

Several studies have pointed out that the CLIP family's vision representation "lacks visual details" (Lu et al., 2024a; Tong et al., 2024b; Ye et al., 2024). We explain this observation through the concept of correspondence. Current multi-modal large language models (MLLMs) convert images into embeddings, with each embedding representing a patch of the image. Image features with high correspondence increase the similarity within internal image patches on similar semantics, thereby enabling the retrieval of more detailed information.

# 3 LAW OF VISION REPRESENTATION IN MLLMS

We introduce the Law of Vision Representation in Multimodal Large Language Models (MLLMs). It states that the performance of a MLLM, denoted as $Z$, can be estimated by two factors: cross-modal alignment ($A$) and correspondence ($C$) of the vision representation, assuming vision representation is the sole independent variable while other components (*e.g.,* language model and alignment module) remain fixed. This relationship can be expressed as:

$$Z \propto f(A, C) \tag{1}$$

where $f$ is a linear function on second-degree polynomial transformations of $A$ and $C$.

## 3.1 ASSUMPTIONS

Following NVLM (Dai et al., 2024), we categorize MLLMs into the following types: (1) Decoder-only MLLMs (Tong et al., 2024a; Liu et al., 2024e; Li et al., 2024a; Liu et al., 2024f; Dai et al., 2024; Lu et al., 2024b; Zhang et al., 2024a; Wang et al., 2024a): These MLLMs consist of vision encoder(s) and an alignment module, such as a multilayer perceptron (MLP), which maps the vision representation into vision tokens. These tokens are designed to have a similar distribution as language tokens and are directly input into a language model in the same manner as language tokens. (2) Cross-attention-based MLLMs (Dai et al., 2024; Bai et al., 2023; Alayrac et al., 2022; Laurençon et al., 2024; Chen et al., 2024c): These MLLMs include vision encoder(s) and an additional module, often serving as a downsampling component, such as a perceiver resampler. The vision tokens generated are integrated into the language model through cross-attention mechanisms.

- The Law of Vision Representation specifically focuses on decoder-only MLLM architecture due to their widespread adoption and their simplicity, which facilitates controlling variables in training recipes and enables clear mathematical modeling.
- We further assume vision representation is the only independent variable, while the alignment module and LLM architecture remain fixed. In the case of a unfrozen vision encoder, we cannot guarantee that the vision encoder does not take the function of the alignment module. This causes the architecture and role of the alignment module to change alongside the encoder, making the experiment uncontrolled and the models no longer comparable.

## 3.2 THEORETICAL JUSTIFICATION

In this section, we theoretically analyze how an increase in $A$ and $C$ leads to improved model performance. When a vision representation demonstrates high cross-modal alignment and accurate correspondence, the MLLM exhibits the following desired properties:

- *When training a MLLM, if the vision representation is closely pre-aligned with the language distribution, the pretrained language model requires less computational effort to bridge the gap between different modalities during finetuning.* In Section A.1, we provide theoretical justification that finetuning on well-aligned multimodal data is about equivalent to finetuning on text-only data, eliminating additional effort beyond language finetuning. This efficiency can lead to improved performance, especially in scenarios where the available training data for finetuning is limited.
- *If the vision representation ensures accurate correspondence, the attention within the image embeddings is precise.* Consequently, the MLLM develops a refined focus on visual content, capturing even details that cannot be derived solely from text-to-image attention, leading to a more detailed interpretation of the image. We provide theoretical justification in Section A.2.

## 3.3 EMPIRICAL JUSTIFICATION

In this section, we empirically show that AC is strongly correlated to model performance. To quantify the correlation between AC and model performance, we first propose methods to measure cross-modal alignment and correspondence within the vision representation:

- To quantify cross-modal alignment, we aim to compare the image and text embeddings of the same concept. However, finding the same concept is difficult since it requires alignment. To address this, we use the CLIP vision embedding as a reference. We calculate the maximum cosine similarity $S_C$ between vector pairs from the CLIP embedding $\hat{E}$ and target vision representation embedding $E$:

$$\text{A SCORE} = \frac{1}{n} \sum_{i=1}^{n} \frac{1}{|E_i|} \sum_{v=1}^{|E_i|} \max_{u} S_C(\hat{E}_i^{(u)}, E_i^{(v)}) \tag{2}$$

where $n$ is the total number of image samples, $|E_i|$ is the number of vectors in the embedding $E_i$, and $E_i^{(v)} = MLP(F_i)^{(v)}$ is the $v$-th embedding vector, resulting from the vision feature $F$ of the $i$-th image.

- To compute the correspondence score, we extract features from $n$ pairs of images, resulting in $F_i^s$ and $F_i^t$ from the $i$-th source and target image pair. Given ground truth key points $\{p_{i1}^s, \ldots, p_{im}^s\}$, we obtain the predicted key points $\{p_{i1}^t, \ldots, p_{im}^t\}$ using the features. The correspondence score is the Percentage of Correct Keypoints (PCK) calculated using the following equation:

$$\text{C SCORE} = \frac{1}{n} \sum_{i=1}^{n} \frac{1}{m} \sum_{j=0}^{m} \mathbb{1}_{\left\|p_{ij}^t - p_{ij}^s\right\|_2 < T} \tag{3}$$

where $T$ is a threshold defined as proportional to the bounding box size of the object instance in the image.

Finally, AC score is a second-degree polynomial transformation of the $A$ and $C$ score:

$$\text{AC SCORE} = \sum_{\alpha=0}^{2} \sum_{\beta=0}^{2-\alpha} w_{\alpha\beta} A^{\alpha} C^{\beta} \tag{4}$$

**Results.** We fit a simple linear regression model using 13 vision representations across 4 vision-based MLLM benchmarks. As shown in Figure 2, the average coefficient of determination ($R^2$) obtained is 95.72% when using the AC score of the vision representations. For comparison, we also fit models using 13 random scores, the A score alone, and the C score alone, all with polynomial transformations. The random scores and single-factor models show significantly lower correlations with performance. This result highlights *the strong correlation between the AC score and MLLM performance, validating the Law of Vision Representation*. Refer to Section 5.4 for details.

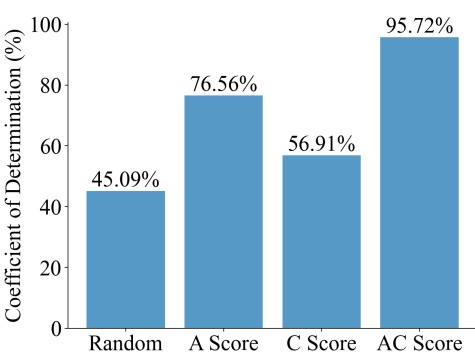

Figure 2: $R^2$ values for linear regression models fitted on various scores.

## 4 AC POLICY

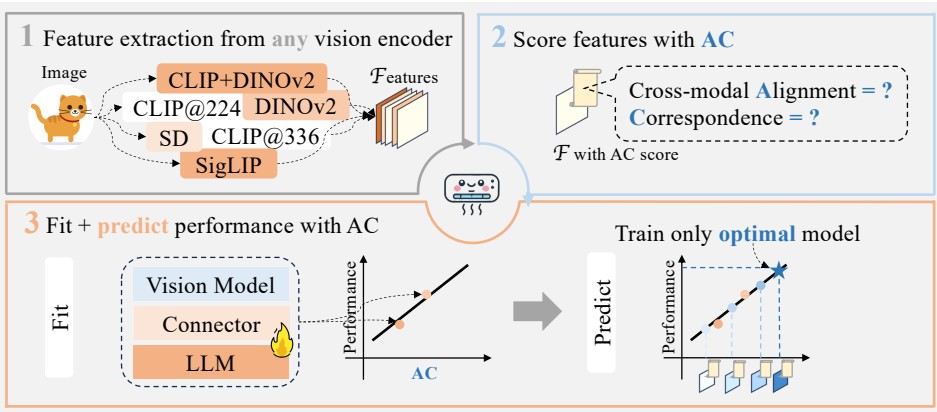

Figure 3: Overall framework of AC policy.

**Problem Formulation.** The MLLM architecture assumed in this framework consists of a frozen vision encoder, followed by a trainable connector (alignment module) and the language model. To determine the optimal out of $k$ vision representations, we originally needs finetune LLM $k$ times, making the scaling of $k$ difficult. Therefore, we propose AC policy, as illustrated in Figure 3, to efficiently estimate the optimal vision representation from a search space consisting of $k$ out of all $2^N - 1$ possible vision representations, given $N$ vision encoders. We finetune only $k'$ LLMs to obtain downstream performance, allowing $k$ to scale without significant cost, where $k' \ll k$. The value of $k'$ is determined based on the computational budget allocated for vision representation selection.

**Policy Fitting.** Let $\mathbf{X} \in \mathbb{R}^{k \times 6}$ be the matrix containing AC scores of vision representation in the search space. We subsample $k'$ data points from $\mathbf{X}$, denoted as $\mathbf{X}_s \in \mathbb{R}^{k' \times 6}$, to serve as the input to the linear regression model:

$$\mathbf{y} = \mathbf{X}_s \mathbf{w} + \epsilon \tag{5}$$

Here, $\mathbf{w} \in \mathbb{R}^6$ is the vector of model parameters, $\epsilon \in \mathbb{R}^{k'}$ is the vector of error terms, and $\mathbf{y} \in \mathbb{R}^{k'}$ represents the downstream performance on a desired benchmark.

**Sampling Strategy.** The selection of $k'$ can impacts the function fit and, consequently, the accuracy of predictions. To avoid sampling points that are too close in terms of their A and C scores, we employ a sampling strategy based on the coordinates.

The normalized A and C score pairs of $k$ vision representation can be plotted on a 2D graph as coordinates $(A, C)$, To ensure diverse sampling, we divide the graph into regions. For each iteration $j$ in which the total sampled points do not yet fulfill $k'$, we divide the graph into $4^j$ equal regions. We then remove empty regions and those that contain previously sampled points. The next data point is randomly selected from a remaining region.

**Results.** In Table 1, we show that *AC policy consistently predicts the optimal vision representation with minimal resources*, given a finite search space—in this case, 13 settings. Our goal is to finetune only a small subset of the search space while still identifying the optimal vision representation within the top-3 predictions (Recall@3). However, if we randomly select a subset to train on, we need 12 out of 13 finetuning to achieve over 90% Recall@3. In contrast, the AC policy requires only 3.88 full training runs on average to reach 89.69% Recall@3. Refer to Section 5.5 for details.

| Benchmark | Number of Finetuning | Recall@3 |
|---|---|---|
| Random | 12 | 92.04% |
| MMBench | 4 | 90.1% |
| MME | 3 | 90.6% |
| OKVQA | 3 | 90.0% |
| SEED-Bench | 3 | 86.0% |
| MMMU | 7 | 96.5% |
| TextVQA | 5 | 83.1% |
| VizWiz | 3 | 84.6% |
| ScienceQA | 3 | 96.6% |
| Average | **3.88** | 89.69% |

Table 1: Number of LLM finetunings required to achieve approximately 90% Recall@3 in predicting the optimal vision representation.

# 5 EMPIRICAL RESULT DETAILS

## 5.1 EXPERIMENT SETTINGS

For our specific MLLM pipeline, we follow the training procedure, general architecture, and dataset outlined in LLaVA (Liu et al., 2024e). The training process consists of two stages: in the first stage, we train a 2-layer GeLU-MLP connector using the LLaVA 1.5 dataset with 558K samples. In the second stage, we train both the connector and the language model, Vicuna-7B 1.5 (Zheng et al., 2023), on the expanded LLaVA 1.5 dataset with 665K samples. We also conduct experiments on LLMs with different size and type shown in Section A.7. It is important to note that for each training, all factors are held constant except for the vision representation being changed. The 13 vision representation in this paper are outlined in Table 2. The MLLM benchmarks used in this pa-

| Vision Representation | Resolution |
|---|---|
| *Single vision encoder: feed-forward models* | |
| OpenAI CLIP ViT-L/14 | 224 |
| OpenAI CLIP ViT-L/14 (Radford et al., 2021) | 336 |
| OpenCLIP ViT-L/14 (Cherti et al., 2023) | 224 |
| SigLIP ViT-L/16 (Zhai et al., 2023b) | 224 |
| DINOv2 ViT-L/14 (Oquab et al., 2023) | 224 |
| *Single vision encoder: diffusion models* | |
| SD 1.5 (Rombach et al., 2022) | 768 |
| SD 2.1 (Rombach et al., 2022) | 768 |
| SD Image Variations | 768 |
| SD XL (Podell et al., 2023) | 512 |
| DiT (Peebles & Xie, 2023) | 512 |
| SD 3 (Esser et al., 2024) | 512 |
| *Multiple vision encoders: feature combination* | |
| CLIP+DINOv2 ViT-L/14 | 224 |
| CLIP+DINOv2 ViT-L/14 | 336 |

Table 2: Vision representation explored.

per includes 4 vision-based benchmarks, MMBench (Liu et al., 2023), MME (Fu et al., 2023), OKVQA (Marino et al., 2019), SEED-Bench (Li et al., 2024b), and 4 QCR-based benchmarks including, MMMU (Yue et al., 2024), TextVQA (Singh et al., 2019), VizWiz (Gurari et al., 2018), ScienceQA (Lu et al., 2022).

## 5.2 AC SCORE

To compute the cross-modal alignment score, we perform stage 1 training with all the vision representations to obtain the MLPs. This process requires significantly less computation than stage 2, involving only 0.298% of the trainable parameters. The alignment score for each benchmark is averaged across 100 randomly sampled images. For the correspondence score, we follow common practices using the SPair-71k (Min et al., 2019) dataset. Consequently, each benchmark has its own alignment score, while the correspondence score remains consistent across all representations.

## 5.3 FEATURE EXTRACTION

Both MLLM training and score computation involve image feature extraction. Below, we introduce the approach for obtaining two types of vision representations.

**Vision Representation from Feed-forward Models.** Given an image $I \in \mathbb{R}^{H \times W \times 3}$ we process it either in its raw form for U-Net models or in a patchified form for transformer models. For transformers, we extract the last hidden state $F \in \mathbb{R}^{l \times c}$ where $l$ is the sequence length and $c$ is the hidden dimension. In the case of the U-Net model, we take the intermediate activation $F \in \mathbb{R}^{\hat{H} \times \hat{W} \times c}$ after the first upsampling block. Note that the features from these two types of models are interchangeable between sequence and grid formats through reshaping and flattening. For consistency, the following sections assume that all features have been pre-converted into the same format.

**Vision Representation from Diffusion Models.** Diffusion model is primarily used for generating images via multi-step denoising, yet a recent trend is to use diffusion model as the vision representation model (Xu et al., 2024; 2023; Zhang et al., 2024c; Tong et al., 2024a). Specifically, for diffusion models, given an image $I \in \mathbb{R}^{H \times W \times 3}$, we first add noise to the VAE-encoded representation of $I$:

$$x_t = \sqrt{a_t} \cdot \text{VAE}(I) + (\sqrt{1 - a_t}) \cdot \epsilon \tag{6}$$

where $\epsilon \sim \mathcal{N}(0, \mathbf{I})$ and $a_t$ is determined by the noise schedule. Note that we utilize the little-noise strategy by setting the $t = 1$. In that case, the diffusion model only denoises the noise-latents once and we treat the one-step denoising latents as the vision representation features.

## 5.4 ADDITIONAL RESULTS ON THE LAW OF VISION REPRESENTATION

In Section 3, we demonstrate the strong correlation between the AC score and MLLM performance by analyzing the coefficient of determination ($R^2$) obtained from fitting a linear regression model. In this section, we further ablate the experiments by adding baselines, fitting model performance with random scores, A scores, and C scores separately. Additionally, we explored the relationship between the A score and C score by applying two different data transformations: no transformation and second-degree polynomial transformation. We avoid higher-degree transformations to prevent overfitting, which could obscure the true relationship between A and C scores.

As shown in Table 3, the results indicate that using the AC score consistently outperforms all other settings in terms of $R^2$ values. While this observation holds regardless of transformation, applying a second-degree polynomial

| Fitting Data | $R^2$ (Vision) | $R^2$ (OCR) |
|---|---|---|
| *No transformation on fitting data* | | |
| Random | 2.94% | 5.08% |
| A Score | 50.80% | 55.06% |
| C Score | 46.12% | 19.45% |
| AC Score | 83.24% | 66.18% |
| *Polynomial transformation on fitting data* | | |
| Random | 45.09% | 35.77% |
| A Score | 76.56% | 77.45% |
| C Score | 56.91% | 29.07% |
| AC Score | 95.72% | 85.21% |

Table 3: Averaged $R^2$ results of AC and other baselines fitting on MLLM benchmarks.

transformation to the A and C scores yields the highest correlation with model performance. This suggests an inherent trade-off between A and C scores: vision representations with high cross-modal alignment often exhibit lower correspondence, and vice versa.

Interestingly, we observe a lower correlation between OCR-based benchmark performance and C scores, which leads to a reduced correlation between the AC score and OCR-based benchmark performance. In Section 7.2, we discuss how the use of the SPair-71k correspondence dataset across all benchmarks fails to adequately capture correspondence in images containing text.

## 5.5 ADDITIONAL RESULTS ON THE AC POLICY

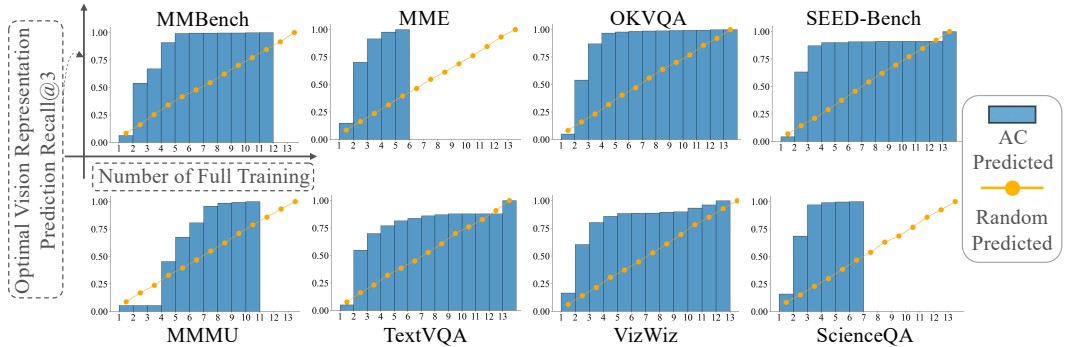

Figure 4: Number of full training (LLM finetuning) cycles required to include the optimal vision representation within the top-3 predictions (Recall@3).

In Section 4, we demonstrate that fitting the AC score consistently predicts the optimal vision representation with minimal resources, given a finite search space—in this case, 13 settings. In this section, we provide detailed visualization for Table 1.

When performing ablation experiments on vision encoders, it's common to randomly select a subset to train on. However, as shown in Figure 4, with 1000 runs of simulated ablation experiments, we found that to include the optimal vision representation 81.2% of the time, at least 11 out of the 13 settings need to be trained. This suggests that running a small subset of vision representations is unreliable, especially as the search space expands, making it increasingly unlikely to identify the true optimal representation by training only a subset.

In contrast, the AC policy requires only 3.88 full training runs on average to reach 89.69% Recall@3. For the most successful prediction benchmark, ScienceQA, the policy successfully identifies the optimal configuration among the top three choices in 96.6% of cases, with only three language model finetuning runs across a 13-setting search space. This result shows that AC policy significantly reduces the effort and cost of exploring vision representations for MLLMs.

## 6 DISCUSSION

### 6.1 FINDING VISION REPRESENTATIONS WITH HIGH AC

Since AC score is highly correlated with MLLM benchmark performance, to improve MLLM from the vision side, it is essential to identify vision representations with high AC scores and add them into the search space. We suggest two strategies to achieve this: increasing resolution and combining features.

Figure 5 shows the normalized performance improvement summed across eight benchmarks. Increasing the resolution of well-aligned features directly enhances correspondence. For example, increasing the image resolution from 224 to 336 for CLIP, while maintaining cross-modal alignment, resulted in a performance increase from 7.1 to 7.3 out of 8.

Additionally, feature combination—merging two features with high A and C scores along the channel dimension—can enhance cross-alignment while preserving correspondence. We chose to con-

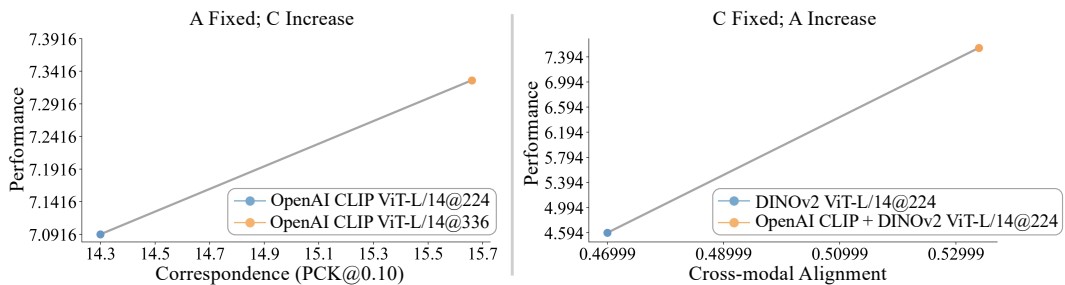

Figure 5: Normalized performance improvements across 8 benchmarks by increasing resolution and combining features, thereby enhancing correspondence and cross-modal alignment.

catenate features along the channel dimension to preserve context length and to align with the intuition from our high correspondence attention proof in Appendix A.2. Specifically, the high correspondence feature can support the retrieval of information in the attention mechanism for the feature with high cross-modal alignment. Combining CLIP with DINOv2, for instance, leads to an increase in cross-modal alignment, thereby improving model performance.

## 6.2 ANALYSIS OF POLICY FITTING COEFFICIENTS

| Benchmark | Fitted Equation |
|---|---|
| MMBench | $-0.127 + 3.191A + 0.910C - 0.721A^2 - 1.570A \cdot C - 0.189C^2$ |
| MME | $-0.163 + 1.915A + 1.591C - 1.318A^2 - 0.073A \cdot C - 1.075C^2$ |
| OKVQA | $-0.121 + 2.367A + 0.933C - 1.804A^2 + 0.156A \cdot C - 0.503C^2$ |
| SEED-Bench | $0.938 + 1.740A + 1.345C - 1.271A^2 + 0.322A \cdot C - 0.856C^2$ |
| MMMU | $-0.241 + 2.747A + 1.416C - 2.145A^2 - 0.242A \cdot C - 1.212C^2$ |
| TextVQA | $-0.065 + 0.842A + 0.756C - 1.519A^2 + 2.943A \cdot C - 1.062C^2$ |
| VizWiz | $0.069 + 2.145A - 0.102C - 1.772A^2 + 0.720A \cdot C + 0.116C^2$ |
| ScienceQA | $0.013 - 0.198A + 1.422C - 0.487A^2 + 2.400A \cdot C - 1.409C^2$ |

Table 4: Fitted equations for each benchmark. The equations describe the policy fitted to performance, where $A$ and $C$ represent specific variables contributing to the benchmarks.

- Positive coefficient for $A$ and $C$ terms: As A and C scores increase, benchmark performance increases proportionally to the coefficients of these terms.
- Negative coefficient for $A^2$ and $C^2$ terms: This indicates a diminishing return or saturation effect—while increasing A and C initially improves performance, their positive impact decreases as the scores grow larger.
- Positive $A$ and $C$ interaction term ($A \cdot C$): A positive coefficient for the interaction term implies that simultaneous increases in A and C magnify their combined effect on performance, resulting in a synergistic boost.

Comparing across benchmarks, we observe that for vision-based benchmarks (MMBench, MME, OKVQA, SEED-Bench), the coefficients for A and C terms are consistently positive, indicating that cross-modal alignment and correspondence are positively correlated with performance. They all have a heavier weight on A than C. However, we also note that the effect diminishes as A and C scores continue to increase, eventually plateauing (at the maximum score for the benchmark). This is due to the negative coefficients for the square terms, which represent a saturation effect where performance gains slow down as A and C scores grow.

For OCR-based benchmarks, such as VizWiz and ScienceQA, we observe unexpected coefficients. We attribute this to the issue discussed in Section 7.2. Specifically, our correspondence calculation image set is based on natural images rather than images containing text, which leads to inaccuracies in measuring correspondence for these tasks. As a result, the fitted function is less effective for OCR-based benchmarks.

# 7 LIMITATIONS

## 7.1 REFINING A SCORE DESIGN

We design the A score calculation as the maximum cosine similarity between each pair of embedding vectors from the CLIP embedding and the target embedding. However, correspondence effects and other bias can be unintentionally included if the features obtained from CLIP and the target encoder differ in resolution. For example, the A score computed for CLIP@224 with CLIP@336 is not the same, although they share the same architecture. This shows that correspondence is not fully disentangled in the A score calculation in this scenario. The best practice is to always use the same input resolution for both CLIP and the target encoder.

Another limitation is the use of CLIP as a reference metric. This could be problematic if another encoder with better cross-modal alignment exists in the search space. However, we believe the error should not be significant enough to cause the predicted optimal vision representation to fall outside the top-3 or top-5. Our main purpose of this work is to model the relationship between performance, alignment, and correspondence, so that we use A score as an initial approach to quantify the concept of alignment in the field of MLLM. A score is designed to be flexible and can be adjusted based on individual preferences. In practice, we use an average of A score from both CLIP@224 and CLIP@336, aiming to average out the influence of correspondence.

## 7.2 REFINING C SCORE DESIGN

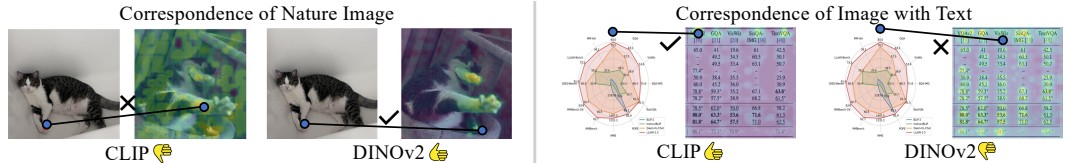

Figure 6: Visualization of correspondence on natural images and images containing text for CLIP and DINOv2. The left image shows the source image, while the right one shows the target image. Blue dots indicate key points with the highest similarity, and the green areas represent regions with relatively high similarity.

We have noted that OCR-based benchmark performance shows a weaker correlation with the AC score compared to vision-based benchmarks. The primary cause of this discrepancy lies in the correspondence dataset we selected to compute the C score. The SPair-71k dataset measures feature correspondence for natural images, such as objects like cats and trains. In Figure 6, the CLIP encoder demonstrates poorer correspondence in natural images compared to DINOv2. However, when it comes to images containing text, CLIP exhibits significantly better correspondence than DINOv2 or any other encoders. Therefore, using SPair-71k to calculate the C score does not accurately capture true correspondence across all scenarios. Ideally, each benchmark should have its own keypoint-labeled images for correspondence evaluation. At a minimum, an OCR-specific correspondence dataset would be highly beneficial for assessing MLLMs. To our knowledge, no such dataset currently exists. We encourage further investigation in this direction, as it would be valuable across fields in MLLM, particularly for understanding tables and charts—a fundamental capability.

# 8 CONCLUSION

In conclusion, we introduce the Law of Vision Representation for decoder-only MLLMs, highlighting the strong correlation between cross-modal alignment, vision representation correspondence, and MLLM performance. Using the AC score to quantify these factors, we demonstrate its linear relationship with performance across extensive experiments. Its application, AC policy, enables the efficient identification and training of the optimal vision representation without repeated fine-tuning of the language model, achieving a 99.7% reduction in computational cost.

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

## A APPENDIX

### A.1 THEORETICAL JUSTIFICATION OF VISION REPRESENTATION WITH HIGH CROSS-MODAL ALIGNMENT

In Section 3.2, we state that when training an MLLM, if the vision representation is closely pre-aligned with the language distribution, then the pretrained language model requires less computational effort to bridge the gap between different modalities during finetuning. In this section, we show that using well-aligned vision representation, finetuning on multimodal data is about equivalent to finetuning on text-only data, eliminating additional effort beyond language finetuning.

Assume the vision embedding distribution $D_{image}$ and text embedding distribution $D_{text}$ are well-aligned in the MLLM. For a shared concept $c$, the image embedding after the alignment module and its corresponding text embedding, $E_c^{image} \sim D_{image}$ and $E_c^{text} \sim D_{text}$, are close in distance, meaning:

$$\|E_c^{image} - E_c^{text}\| \leq \epsilon \tag{7}$$

where $\epsilon$ is a small constant. Given this condition, we can show that the output of the MLLM with multimodal embeddings $[E_c^{image}, E_1, E_2, \ldots, E_n]$ is close to the output with text-only embeddings $[E_c^{text}, E_1, E_2, \ldots, E_n]$.

Since our language model $f$ is well-trained and pre-normed, the input space to each transformer layer is bounded and compact, meaning that the values of the input are bounded by $c$, a small constant. This implies that the continuously differentiable function $f$ is Lipschitz (Kim et al., 2021). This property ensures that small changes in the input of the language model of the MLLM result in small, controlled changes in the output:

$$\|f([E_c^{\text{image}}, E_1, E_2, \ldots, E_n]) - f([E_c^{\text{text}}, E_1, E_2, \ldots, E_n])\| \leq L\|[E_c^{\text{image}}, E_1, E_2, \ldots, E_n] \tag{8}$$

$$- [E_c^{\text{text}}, E_1, E_2, \ldots, E_n]\| \tag{9}$$

$$\leq L\epsilon \tag{10}$$

where $L$ is the Lipschitz constant. This closeness in output distance implies that even with multimodal data, the pretrained language model mimics the training dynamics closely resemble language-only finetuning.

## A.2 THEORETICAL JUSTIFICATION OF VISION REPRESENTATION WITH ACCURATE CORRESPONDENCE

In Section 3.2, we state that if the vision representation ensures accurate correspondence, the attention within the image embedding is precise. In this section, we show that vision representation with accurate correspondence can help vision information retrieval in the attention mechanism. Therefore, more visual details are considered even if not attended by the text token.

Consider an input $[E_0^{\text{image}}, E_1^{\text{image}}, E_2, \ldots, E_n]$ to the transformer, where the image embeddings $E_0^{\text{image}}$ and $E_1^{\text{image}}$ are derived from different patch of a high correspondence vision representation. By definition, the dot product $E_0^{\text{image}} \cdot E_1^{\text{image}}$ is large if the two corresponding original image patches share related information.

Suppose a text token $E_2$ attends to $E_0^{\text{image}}$. We show that it is also able to retrieve $E_1^{\text{image}}$ and vice versa. This can be demonstrated as follows:

$$\text{score}(E_2, E_0^{\text{image}}) = \frac{(E_2 W^Q) \cdot (E_0^{\text{image}} W^K)}{\sqrt{d_k}} \tag{11}$$

If $\text{score}(E_2, E_0^{\text{image}})$ is high, and $(E_0^{\text{image}} W^K)^\top (E_1^{\text{image}} W^K)$ is also large (assuming $W^K$ does not distort the vectors drastically), then by transitivity, $\text{score}(E_2, E_1^{\text{image}})$ is also likely to be high. This transitivity ensures that attention is effectively spread across related visual information, enhancing the model's ability to interpret visual content in greater detail.

## A.3 ALL SETTINGS BENCHMARK PERFORMANCE

In this section, we present the performance results of all 13 vision representation settings, as summarized in Table 5. The benchmarks we evaluated include:

- MMBench (Liu et al., 2023): A set of multiple-choice questions designed to assess 20 different ability dimensions related to perception and reasoning.
- MME (Fu et al., 2023): A dataset focused on yes/no questions, covering areas such as existence, counting, position, and color, primarily based on natural images.
- MMMU (Yue et al., 2024): Multiple-choice questions targeting college-level subject knowledge and deliberate reasoning, primarily testing the language model's abilities.
- OKVQA (Marino et al., 2019): Open-ended questions based on the MSCOCO (Lin et al., 2014) dataset, spanning 10 different knowledge categories.
- TextVQA (Singh et al., 2019): Open-ended questions designed to evaluate the model's OCR capabilities.
- VizWiz (Gurari et al., 2018): Open-ended questions sourced from people who are blind, aimed at testing the model's OCR capabilities.
- ScienceQA (Lu et al., 2022): A multiple-choice science question dataset, with 86% of the images being non-natural, covering topics in natural science, social science, and language science.
- SEED-Bench (Li et al., 2024b): A benchmark consisting of multiple-choice questions designed to assess both spatial and temporal understanding.

## A.4 ALL SETTINGS AC SCORES

We provide the AC scores of all 13 vision representation settings, as summarized in Table 6. Additionally, we provide a case analysis comparing the A and C scores of CLIP@224 and CLIP@336 from an intuitive perspective. When comparing these scores, CLIP@224 exhibits a significant drop of 8.7% in the C score compare to CLIP@336, while CLIP@336 shows a slight decrease of 3.0% in the A score. Since CLIP has a high A score, and the effect of A diminishes as it enters the high range as detailed in Section 6.2, this slight drop is expected to have minimal impact on the predicted

| | CLIP@336 | CLIP@224 | OpenCLIP | DINOv2 | SDim | SD1.5 | SDXL |
|---|---|---|---|---|---|---|---|
| MMBench | 64.26 | 64.18 | 63.406 | 58.51 | 52.84 | 42.53 | 43.73 |
| MME | 1502.70 | 1449.64 | 1460.28 | 1295.47 | 1205.33 | 1163.90 | 1212.69 |
| MMMU | 35.0 | 36.2 | 37.2 | 34.6 | 33.7 | 33.9 | 32.8 |
| OKVQA | 53.20 | 56.13 | 56.36 | 54.78 | 46.04 | 39.14 | 41.78 |
| TextVQA | 46.04 | 42.67 | 40.13 | 14.27 | 13.77 | 11.64 | 11.81 |
| VizWiz | 54.27 | 51.69 | 52.11 | 49.69 | 47.33 | 50.14 | 47.14 |
| ScienceQA (Full) | 69.97 | 70.23 | 69.78 | 68.17 | 68.66 | 66.75 | 67.72 |
| ScienceQA | 70.31 | 70.90 | 70.69 | 69.60 | 69.77 | 68.38 | 68.90 |
| ScienceQA (Img) | 69.26 | 68.82 | 67.87 | 65.15 | 66.34 | 63.31 | 65.25 |
| SEED-Bench (Img) | 66.09 | 65.13 | 64.71 | 61.39 | 50.33 | 50.00 | 53.78 |
| SEED-Bench | 60.44 | 60.28 | 59.31 | 57.13 | 46.60 | 46.45 | 49.09 |
| SEED-Bench (Video) | 39.02 | 41.90 | 38.86 | 41.02 | 32.50 | 33.01 | 31.33 |
| | DiT | SD3 | SD2.1 | SigLIP | C+D@224 | C+D@336 | |
| MMBench | 33.68 | 32.82 | 28.87 | 61.86 | 65.72 | 65.12 | |
| MME | 902.00 | 843.43 | 905.27 | 1425.00 | 1436.42 | 1475.19 | |
| MMMU | 32.7 | 32.4 | 32.8 | 35.8 | 36.9 | 34.6 | |
| OKVQA | 33.75 | 34.95 | 34.41 | 54.01 | 55.94 | 56.92 | |
| TextVQA | 10.82 | 10.77 | 10.46 | 36.00 | 40.04 | 46.17 | |
| VizWiz | 49.92 | 47.12 | 46.59 | 53.17 | 54.04 | 53.44 | |
| ScienceQA (Full) | 66.62 | 66.14 | 65.69 | 69.11 | 70.87 | 69.45 | |
| ScienceQA | 68.12 | 67.98 | 67.13 | 70.17 | 71.70 | 70.31 | |
| ScienceQA (Img) | 63.46 | 62.27 | 62.67 | 66.88 | 69.11 | 67.63 | |
| SEED-Bench (Img) | 40.66 | 38.94 | 38.82 | 64.40 | 65.39 | 66.38 | |
| SEED-Bench | 38.31 | 36.96 | 36.84 | 59.41 | 60.47 | 61.39 | |
| SEED-Bench (Video) | 29.41 | 29.46 | 29.33 | 40.48 | 41.84 | 42.48 | |

Table 5: Benchmark performance of all 13 settings. C+D means feature combination of CLIP and DINOv2. The table provides data points for function fitting and is not intended for comparison.

performance. However, we caution against using A and C scores alone to infer performance without fitting a regression function.

The higher C score for CLIP@336 compared to CLIP@224 is reasonable, as the higher resolution better preserves visual details, resulting in stronger correspondence. Conversely, the higher A score for CLIP@224 compared to CLIP@336 reflects an underexplored area: to our knowledge, no prior work has attempted to measure cross-modal alignment in the same contrastive learning model under varying input resolutions. As a result, it is challenging to draw definitive conclusions about the true cross-modality behavior in this case.

## A.5 MORE VISUALIZATION OF CORRESPONDENCE

We provide additional visualizations of correspondence for four different vision representations: CLIP, SigLIP, DINOv2, and Stable Diffusion 1.5. Figures 7 and 8 display pairs of source-target images for each of the four vision representations. In each pair, the left image is the source, and the right image is the target. The red dot on both images indicates the predicted key points using the vision representation. Ideally, these key points should correspond to the same semantic meaning. For example, a red dot on the "left cat ear" in the source image should correspond to the "left cat ear" in the target image. The green areas highlight regions of relatively high similarity with the source points.

In Figure 7, DINOv2 demonstrates superior correspondence for natural images compared to the other vision representations. It accurately matches small parts of the cat between the left and right images, whereas CLIP struggles to correctly identify and align features such as left, right, front, and back.

In Figure 8, , the CLIP family shows precise correspondence for text within images. For instance, when the source image points text like "LLaVA" or "VQAv2", CLIP accurately matches all instances of the text in the target image. In contrast, other vision representations known for "accurate correspondence" in computer vision, such as DINOv2 and Stable Diffusion, fail to provide the same

|  | CLIP@336 | CLIP@224 | OpenCLIP | DINOv2 | SDim | SD1.5 | SDXL |
|---|---|---|---|---|---|---|---|
| *Correspondence* | | | | | | | |
| PCK@0.10 | 15.66 | 14.3 | 16.22 | 24.51 | 20.9 | 22.02 | 16.52 |
| *Cross-modal Alignment* | | | | | | | |
| MMBench | 0.788 | 0.815 | 0.524 | 0.462 | 0.349 | 0.351 | 0.357 |
| MME | 0.791 | 0.818 | 0.532 | 0.472 | 0.370 | 0.366 | 0.358 |
| MMMU | 0.782 | 0.813 | 0.512 | 0.464 | 0.356 | 0.353 | 0.358 |
| OKVQA | 0.801 | 0.825 | 0.543 | 0.472 | 0.379 | 0.375 | 0.353 |
| TextVQA | 0.797 | 0.820 | 0.536 | 0.458 | 0.374 | 0.367 | 0.347 |
| VizWiz | 0.794 | 0.815 | 0.530 | 0.469 | 0.370 | 0.372 | 0.360 |
| ScienceQA (Img) | 0.799 | 0.826 | 0.541 | 0.472 | 0.371 | 0.356 | 0.342 |
| SEED-Bench (Img) | 0.810 | 0.829 | 0.554 | 0.491 | 0.376 | 0.359 | 0.342 |

|  | DiT | SD3 | SD2.1 | SigLIP | C+D@224 | C+D@336 |
|---|---|---|---|---|---|---|
| *Correspondence* | | | | | | |
| PCK@0.10 | 1.91 | 3.09 | 6.99 | 12.89 | 23.62 | 26.08 |
| *Cross-modal Alignment* | | | | | | |
| MMBench | 0.387 | 0.374 | 0.348 | 0.505 | 0.537 | 0.512 |
| MME | 0.392 | 0.377 | 0.338 | 0.526 | 0.536 | 0.511 |
| MMMU | 0.398 | 0.363 | 0.333 | 0.499 | 0.526 | 0.504 |
| OKVQA | 0.366 | 0.390 | 0.351 | 0.540 | 0.537 | 0.514 |
| TextVQA | 0.368 | 0.387 | 0.347 | 0.532 | 0.530 | 0.506 |
| VizWiz | 0.383 | 0.401 | 0.348 | 0.527 | 0.525 | 0.505 |
| ScienceQA (Img) | 0.363 | 0.383 | 0.358 | 0.541 | 0.537 | 0.512 |
| SEED-Bench (Img) | 0.367 | 0.388 | 0.371 | 0.549 | 0.545 | 0.525 |

Table 6: AC scores of all 13 settings. C+D means feature combination of CLIP and DINOv2. The table provides data points for function fitting and is not intended for comparison.

level of accuracy when dealing with images containing text. This emphasizes a key distinction in selecting vision representations for computer vision tasks versus multimodal large language models (MLLMs).

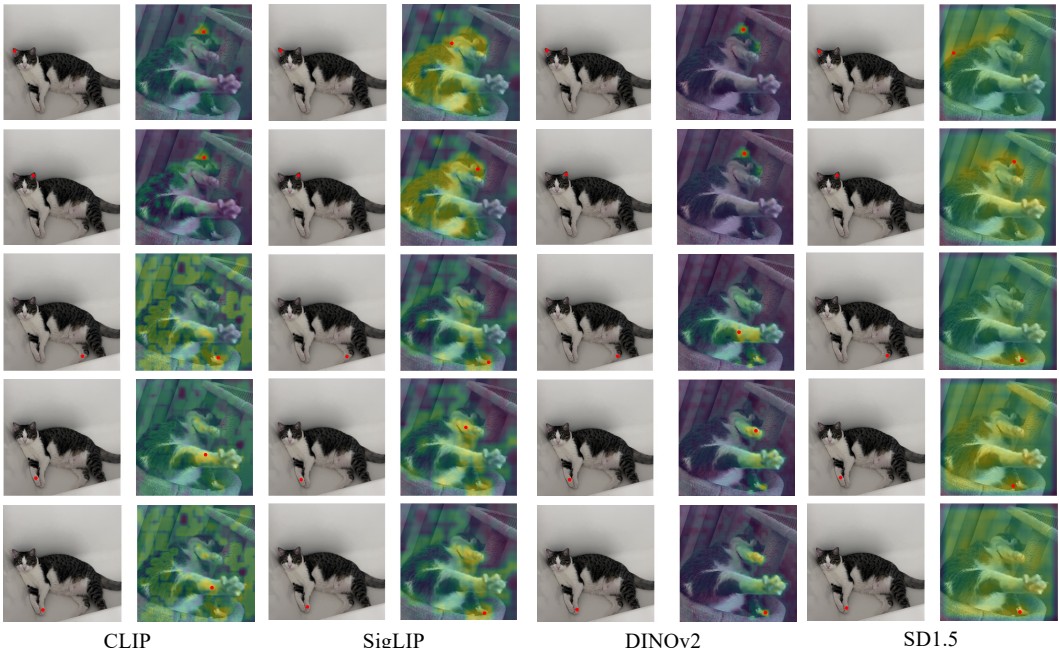

CLIP  SigLIP  DINOv2  SD1.5

Figure 7: Correspondence of natural images for different vision representations.

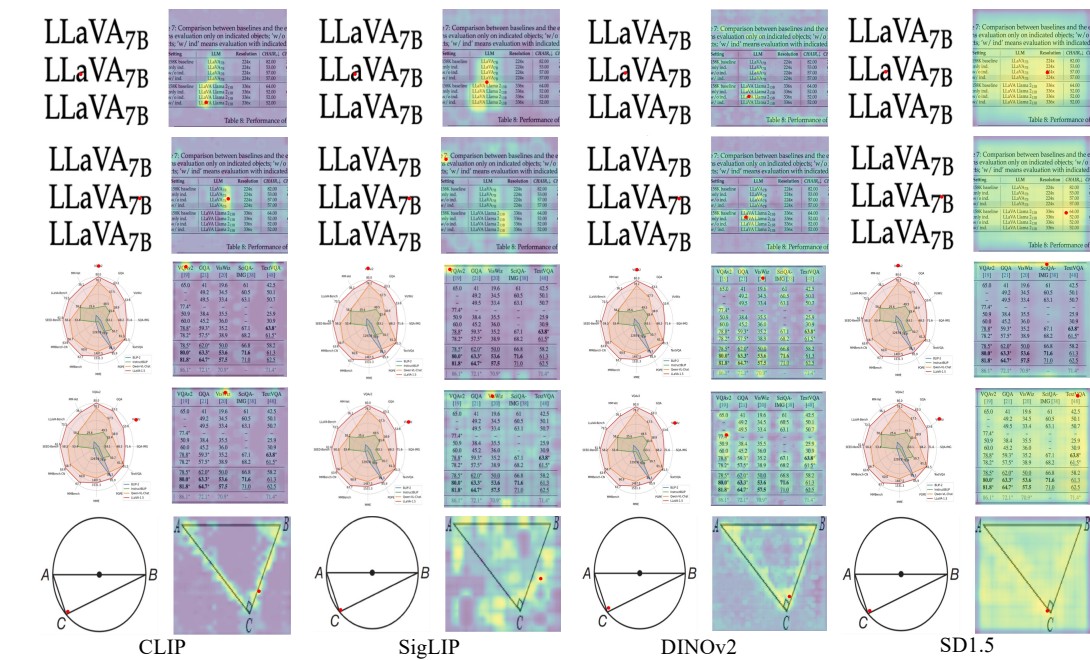

Figure 8: Correspondence of images with text for different vision representations.

### A.6 PSEUDO CODE

---

**Algorithm 1:** COMPUTE A SCORE

---

**Input:** 100 images $I$ from benchmark $B$; vision encoders $CLIP224$, $CLIP336$, and $target$ with pretrained projectors
**Output:** $A$ score for target vision representation on benchmark $B$

```
clip224_tensors ← CLIP224(I);           // Shape: [100, sequence length,
hidden dimension]
clip336_tensors ← CLIP336(I);
target_tensors ← target(I);
cosine_similarities_336 ← [];
cosine_similarities_224 ← [];
for clip336_tensor, clip224_tensor, target_tensor ∈
zip(clip336_tensors, clip224_tensors, target_tensors) do
    clip336_tensor ← normalize_feature(clip336_tensor);
    clip224_tensor ← normalize_feature(clip224_tensor);
    target_tensor ← normalize_feature(target_tensor);
    similarity_336 ← cosine_similarity(clip336_tensor, target_tensor);
    similarity_224 ← cosine_similarity(clip224_tensor, target_tensor);
    max_similarity_336 ← max(similarity_336, dim = 1);
    max_similarity_224 ← max(similarity_224, dim = 1);
    cosine_similarities_336.append(mean(max_similarity_336));
    cosine_similarities_224.append(mean(max_similarity_224));

A_score ← (mean(cosine_similarities_336) + mean(cosine_similarities_224))/2;
```

---

**Algorithm 2:** COMPUTE C SCORE

**Input:** Set of paired images with key points $S$ from SPair-71k, vision encoder $E$, threshold
$threshold$

**Output:** $C$ score for vision encoder $E$

$gt\_correspondences \leftarrow []$ ;          // Ground truth keypoint correspondences
$pred\_correspondences \leftarrow []$ ;           // Predicted keypoint correspondences
**foreach** $(img_1, kp_1, img_2, kp_2) \in S$ **do**
  $image\_tensor\_1 \leftarrow E(img_1)$;
  $image\_tensor\_2 \leftarrow E(img_2)$;
  $sim\_matrix \leftarrow image\_tensor\_1 \cdot image\_tensor\_2^T$ ;    // Compute similarity
   matrix
  $kps\_1\_to\_2 \leftarrow$ calculate_keypoint_transformation$(sim\_matrix, kp_1)$ ;    // Transform
   keypoints from $img_1$ to $img_2$
  $gt\_correspondences$.append$(kp_2)$;
  $pred\_correspondences$.append$(kps\_1\_to\_2)$;

$error \leftarrow$ Euclidean_distance$(pred\_correspondences, gt\_correspondences)$;
$correct \leftarrow$ sum$(error < threshold)$;
$C\_score \leftarrow correct/$total keypoints in $kp_2$;

---

**Algorithm 3:** REGION-BASED SAMPLING

**Input:** $k$ A and C score pairs from models $ACs$; $past\_sampled$ models; current sampling $level$
(1 to $k'$, increments when regions are exhausted as each region is sampled only once)

**Output:** Sampled $model$ to train next

$regions \leftarrow \{\}$;
**for** $AC \in ACs$ **do**
  $region\_key \leftarrow$ determine_region$(A, C, level)$ ;  // Identify the region based
   on A and C coordinates
  $regions[region\_key].append((model, A, C))$;

Remove models in $past\_sampled$ from $regions$;
$remaining\_regions \leftarrow$ keys of $regions$;
$chosen\_region \leftarrow$ randomly select from $remaining\_regions$;
$model \leftarrow$ randomly select from $regions[chosen\_region]$;

---

**Algorithm 4:** AC POLICY

**Input:** $k$ vision encoders with pretrained projectors $V$; computation budget $k'$
**Output:** A ranking of $k$ MLLMs based on performance
$ACs \leftarrow [(\text{Compute\_A\_Score}(v), \text{Compute\_C\_Score}(v)) \,|\, v \in V]$;
$past\_sampled \leftarrow []$;
$train\_ACs \leftarrow []$;
$train\_performance \leftarrow []$;
**for** $i \leftarrow 1$ **to** $k'$ **do**
  $model \leftarrow$ Region_based_Sampling$(ACs, past\_sampled)$;
  $performance \leftarrow$ Fully train $model$;
  $train\_ACs$.append(AC of $model$);
  $train\_performance$.append($performance$);
  $past\_sampled$.append($model$);

$poly \leftarrow$ PolynomialFeatures$(degree = 2)$;
$transformed\_train\_ACs \leftarrow poly$.fit_transform$(train\_ACs)$;
$regression \leftarrow$ LinearRegression$()$;
$regression$.fit$(transformed\_train\_ACs, train\_performance)$;
$ranking \leftarrow$ Rank $V$ by regression predictions on $ACs$;

## A.7 FITTING THE LAW OF VISION REPRESENTATION ON MLLM WITH DIFFERENT LLMs

In this paper, we demonstrate the relationship between the AC score and performance by fitting a regression model and reporting the $R^2$ value. As shown in Table 7, the $R^2$ achieves 95.72% when averaged across four vision benchmarks under the LLaVA 1.5 setting, where the LLM is Vicuna-7B 1.5.

| LLM | $R^2$ (Vision) | $R^2$ (OCR) |
|---|---|---|
| Vicuna-7B 1.5 | 95.72% | 85.21% |
| Llama2-7B | 98.01% | 87.91% |
| Vicuna-13B 1.5 | 95.17% | 88.50% |

Table 7: Averaged $R^2$ values for AC score fitting across vision and OCR benchmarks using different LLMs.

In additional experiments, we show that the fitting $R^2$ remains strong, and in some cases even higher, when using different LLM types and sizes. The variation in $R^2$ falls within a reasonable range, indicating that the effect of LLM and vision representation compatibility, if it exists, is negligible compared to the influence of the A and C factors. These results demonstrate that the Law of Vision Representation relationship holds across different LLMs.

## A.8 AC POLICY EVALUATION ON RECALL@1 AND RECALL@2

In Table 1, we demonstrate that, on average, only 3.88 fine-tuning runs are required to achieve 89.69% Recall@3 in identifying the optimal vision representation. This means that, with a computation budget of at most 7 runs, we can find the optimal vision representation in almost 90% of the time. Using this same budget, we further evaluate other metrics, such as Recall@1 and Recall@2, to provide a comprehensive assessment.

| Metric | Average Finetuning Runs | Total Budget | Chance of Finding Optimal VR |
|---|---|---|---|
| Random | 6 | 7 Runs (out of 13) | 48.2% |
| Recall@1 | 6 | 7 Runs | 68.23% |
| Recall@2 | 5 | 7 Runs | 81.01% |
| Recall@3 | 3.88 | 6.88 Runs | 89.68% |

Table 8: Comparison of average finetuning runs and chances of finding the optimal vision representation under different metrics.

We observe in Table 8 that, given the same computation budget where only 7 of the vision representations in the search space can be tested, using Recall@1 provides a significant 20% improvement over random trials in the chance of finding the optimal vision representation. However, using Recall@3 as a metric further optimizes this chance, demonstrating its practicality. This observation implies that, rather than assuming an absolute "optimal vision representation," we should acknowledge that each fine-tuning and inference process is subject to fluctuations, making "optimal" predictions inherently challenging. Employing Recall@3 accounts for these fluctuations, providing a more robust and practical approach without increasing computation costs.

Additionally, we provide the average performance difference between the Top-1 and Top-2 vision representations trained MLLM in Table 9 to illustrate that, in most cases, the performance differences are minimal and their order can be attributed to fluctuations.

| Benchmark | Percentage Difference Between Top-1 and Top-2 VR |
|---|---|
| MMBench | 0.915% |
| MME | 1.83% |
| OKVQA | 1.39% |
| Seed-Bench | 0.434% |
| MMMU | 0.806% |
| TextVQA | 0.269% |
| VizWiz | 0.410% |
| ScienceQA | 0.215% |

Table 9: Percentage performance difference between the Top-1 and Top-2 vision representations across various benchmarks. The differences are minimal, often within a small margin, indicating fluctuations.

