# OpenReview forum: "Law of Vision Representation in MLLMs"
_ICLR.cc/2025/Conference — Submitted to ICLR 2025_

### Official Review · Reviewer_9bVh · 2024-11-02

**Soundness:** 3
**Presentation:** 2
**Contribution:** 2
**Rating:** 5
**Confidence:** 4

**Summary:**

The authors propose the AC score, which combines cross-modal alignment and correspondence in vision representation. Then, they find a linear correlation between the AC score and MLLMs' performance. Extensive experiments across different vision representations validate the linear relationship between the AC score and MLLMs' performance, with an R-squared value of 95.72%.

**Strengths:**

* The motivation is sound. Building the relationship between vision representation and MLLMs' performance is very interesting and helpful.
* By identifying an optimal vision representation without extensive fine-tuning, the paper contributes to reducing computational costs and makes MLLM training more efficient.

**Weaknesses:**

* Cross-modal alignment score is the maximum cosine similarity between each pair of embedding vectors from the CLIP embedding and the target embedding. However, there may be bias in CLIP characterization. At the same time, CLIP is also one of the vision representations involved in the test, which results in the CLIP representation having an artificially high score on the cross-modal alignment score.
* The authors only tested the proposed method on LLaVA v1.5. However, I am concerned that the vision representation law will change after changing the MLLM structure or the base LLMs. Therefore, I strongly recommend the authors to verify the generalization of the proposed method on more MLLM structures or the base LLMs.
* To fairly compare different vision representations, it is usually necessary to control the computation cost (e.g., flops) and model parameters of different vision representations within a similar level, but the proposed method does not take this into account. I suggest providing the computational cost and parameter counts for the different vision representations used and discussing how these factors might impact the AC scores.

**Questions:**

* How would using a stronger contrastive learning model, like BLIP, or a weaker model affect the results when calculating cross-modal alignment score?
* Does calculating the correspondence score require training an MLP to predict key points? If so, different vision representations may require different training hyper-parameters. Consequently, what approach should be taken to determine these optimal hyper-parameters? If the representation dimensions of the vision representation are not the same, then how can they be fairly compared?
* Why does the AC score not exhibit strong performance on the TextVQA and VizWiz benchmarks? Are there any significant differences between these two benchmarks and other benchmarks?
* How is the number of fine-tuning in Tab. 1 determined? Why does the number of fine-tuning vary across different benchmarks?

---

> ### Author Response · Authors · 2024-11-18
> **Reply to Reviewer 9bVh (1/3)**
>
> Thanks for your time in reviewing our work! Here, we address your concerns as following:
>
> ## Weaknesses:
>
> > *W1. Cross-modal alignment score is the maximum cosine similarity between each pair of embedding vectors from the CLIP embedding and the target embedding. However, there may be bias in CLIP characterization. At the same time, CLIP is also one of the vision representations involved in the test, which results in the CLIP representation having an artificially high score on the cross-modal alignment score.*
>
> The reference model for the **A score is designed to be flexible and can be adjusted according to individual preferences**. For example, one can choose a different model as the reference based on the belief that it has the highest cross-modal alignment or to minimize fitting error. So far, we believe that CLIP has the best cross-modal alignment, and it is therefore expected to have the highest score.
> We have also tested SigLIP as the reference embedding, as detailed in our response to Question 1, and found that the results still align with the proposed framework. Importantly, **we are the first to propose and quantify the relationship between cross-modal alignment and MLLM performance, which we believe is already a significant contribution**. We encourage future research to build upon our findings and further improve methods for quantifying cross-modal alignment.
>
> > *W2. Verify the generalization of the proposed method on more MLLM structures or the base LLMs*
>
> In this paper, we demonstrate the relationship between the AC score and performance by fitting a regression model and reporting the $R^2$ value. As shown in paper Table 3, the $R^2$ achieves $95.72$% when averaged across four vision benchmarks under the LLaVA1.5 setting, where the LLM is Vicuna-7B-1.5.
>
> | LLM               | $R^2$ (Vision) | $R^2$ (OCR) |
> |--------------------|---------------------|------------------|
> | Vicuna-7B-1.5     | 95.72%              | 85.21%           |
> | Llama2-7B         | 98.01%              | 87.91%           |
> | Vicuna-13B-1.5    | 95.17%              | 88.50%           |
>
> In additional experiments, we show that the fitting $R^2$ remains strong, and in some cases even higher, when using different LLM types and sizes. The variation in $R^2$ falls within a reasonable range, indicating that the effect of LLM and vision representation compatibility, if it exists, is negligible compared to the influence of the A and C factors. These results demonstrate that **the Law of Vision Representation relationship holds across different LLMs**.
>
> We strongly believe that the Law of Vision Representation applies to a broader range of LLMs. This is because the vision encoder serves as the initial feature extractor for vision information, and changes in the alignment module or LLM do not alter the quality of the information extracted. While the empirical validation of the Law could extend infinitely, finetuning and inference MLLMs is expensive, with the settings tested in this paper and rebuttal costing nearly $100,000. This underscores the significance of the Law of Vision Representation, as it enables others to avoid these high costs. We hope that the additional experiments provided convey the broad applicability of our findings.

---

> ### Author Response · Authors · 2024-11-18
> **Reply to Reviewer 9bVh (2/3)**
>
> > *W3. Providing the computational cost and parameter counts for the different vision representations used and discussing how these factors might impact the AC scores*
>
> | Vision Representation   | FLOPs       | Params      | A Score | C Score | Normalized Average Performance |
> |--------------------------|-------------|-------------|---------|---------|--------------------------------|
> | SDXL                    | 1454.781 G  | 2.56746 G   | 0.352   | 16.52   | 0.309                          |
> | SD3                     | 539.186 G   | 2.02833 G   | 0.383   | 3.09    | 0.030                          |
> | SDim                    | 1370.359 G  | 893.684 M   | 0.368   | 20.9    | 0.399                          |
> | SD1.5                   | 1370.359 G  | 893.684 M   | 0.362   | 22.02   | 0.306                          |
> | SD2.1                   | 1348.904 G  | 900.046 M   | 0.349   | 6.99    | 0.033                          |
> | DiT                     | 537.733 G   | 783.9896 M  | 0.378   | 1.91    | 0.120                          |
> | DINOv2+CLIP336          | 848.011 M   | 607.876 M   | 0.511   | 26.08   | 0.882                          |
> | DINOv2+CLIP224          | 353.357 M   | 607.549 M   | 0.534   | 23.62   | 0.942                          |
> | CLIP336                 | 423.711 M   | 303.507 M   | 0.795   | 15.66   | 0.916                          |
> | DINOv2                  | 176.809 M   | 304.369 M   | 0.47    | 24.51   | 0.574                          |
> | CLIP224                 | 176.548 M   | 303.18 M    | 0.82    | 23.62   | 0.886                          |
> | OpenCLIP                | 176.548 M   | 303.18 M    | 0.534   | 16.22   | 0.892                          |
> | SigLIP                  | 122.286 M   | 92.8842 M   | 0.527   | 12.89   | 0.815                          |
>
>
> The table lists all the vision representations tested in this paper, ranked by FLOPs and parameter counts in descending order. From the results, we observe that **neither the FLOPs nor the parameter count of the vision encoder directly correlates with the AC scores or model performance**. Specifically, larger vision encoders do not necessarily result in better performance or higher AC scores, demonstrating that the size of the vision encoder is not the determining factor for cross-modal alignment or overall model effectiveness.
>
> ## Questions:
>
> > *Q1. How would using a stronger contrastive learning model, like BLIP, or a weaker model affect the results when calculating cross-modal alignment score?*
>
> We explore changing the reference feature used for A score calculation and report the fitting results across four vision benchmarks, both with and without the reference encoder included in the test set.
>
> | Reference VR                      | $R^2$ (Vision) |
> |-----------------------------------|-------------|
> | CLIP with CLIP in the test        | 95.72%      |
> | CLIP without CLIP in the test     | 96.39%      |
> | SigLIP with SigLIP in the test    | 91.99%      |
> | SigLIP without SigLIP in the test | 93.81%      |
>
> The first rows in our analysis correspond to the reported $R^2$ in the paper, demonstrating that when using CLIP, even if CLIP is excluded from the test set (no CLIP in fitting for the second row), the relationship between the AC score and performance remains as strong as when CLIP is in the test set.
>
> We also tested another contrastive learning model, SigLIP (instead of BLIP, primarily due to time constraints), as the reference embedding for A score calculation. The results showed a lower fitting correlation coefficient compared to CLIP, both when SigLIP was included in the fitting set and when excluded. This finding justifies our choice of using CLIP as the reference embedding.
>
> > *Q2. Does calculating the correspondence score require training an MLP to predict key points? If so, different vision representations may require different training hyper-parameters. Consequently, what approach should be taken to determine these optimal hyper-parameters? If the representation dimensions of the vision representation are not the same, then how can they be fairly compared?*
>
> No, we predict key points using an unsupervised method, so no hyperparameters need to be tuned. All vision representations calculate the correspondence score using the same hyperparameters, ensuring fair comparisons across different representations.
>
> For more details on the correspondence score calculation, please refer to the pseudocode provided in the updated Appendix Section 6.

---

> ### Author Response · Authors · 2024-11-18
> **Reply to Reviewer 9bVh (3/3)**
>
> > *Q3. Why does the AC score not exhibit strong performance on the TextVQA and VizWiz benchmarks? Are there any significant differences between these two benchmarks and other benchmarks?*
>
> As discussed in Section 6.3, benchmark performances are less correlated in OCR-based tasks, such as TextVQA and VizWiz, because the correspondence dataset measures the correspondence of natural images rather than text-heavy images. Consequently, the prediction of the optimal vision representation is less accurate for these benchmarks, requiring more finetuning runs to identify the optimal representation.
>
> > *Q4. How is the number of fine-tuning in Tab. 1 determined? Why does the number of fine-tuning vary across different benchmarks?*
>
> The number of finetuning runs in Table 1 is determined to achieve an average Recall@3 of approximately $90$%, which we believe provides the best accuracy for presentation purposes. That being said, Table 1 is only a summary of the data shown in the paper Figure 4, where the full relationship between the number of fine-tuning runs and Recall@3 is detailed.
>
> In practical use, the number of finetuning runs should follow the available computation budget, as explained in our response to Reviewer AFzz’s Weakness 1. The variation in the number of finetuning runs across different benchmarks is due to differences in Recall@3 performance across benchmarks, which is explained in more detail in [our response to Question 3](https://openreview.net/forum?id=SZm3hxmksx&noteId=FZu5Q2HjEb).

---

> ### Author Response · Authors · 2024-11-20
> **Follow-Up on Rebuttal for Your Feedback**
>
> Dear Reviewer 9bVh,
>
> Thank you once again for the time and effort you’ve dedicated to reviewing our work. We have carefully replied to your weaknesses and questions you highlighted in your feedback and provided additional experiments as needed.
>
> We kindly ask if you could let us know whether our responses have sufficiently addressed your concerns. Your continued feedback would be invaluable in helping us further refine the paper during the discussion phase.

---

> > ### Comment · Reviewer_9bVh · 2024-11-22
> >
> > I appreciate the authors' responses. However, I also have some concerns.
> >
> > First, based on my experience and experimental results from related works, using SigLIP in MLLMs should perform better than using CLIP. However, after reviewing the manuscript, I found that the results showed SigLIP MLLMs performed worse than CLIP MLLMs. From the author's responses, it appears that the SigLIP model, which is significantly smaller than the CLIP model, was used for comparison. This raises my first concern: **the experiments presented may not be sufficient to support the paper’s motivation**. Besides, **it seems the author has not demonstrated a model combination that outperforms others in related work (e.g., SigLIP ViT-SO400M/14@384 + DINOv2 ViT-L/14@518 + OpenAI CLIP ViT-L/14@336 + ConvNeXt-XXL@1024)**.
> >
> > Second, the authors' responses reinforce my earlier concerns about the bias of the proposed AC score. Specifically, **the A score is the maximum cosine similarity between each pair of embedding vectors from the CLIP embedding and the target embedding, which results in the CLIP representation having an artificially high score on the cross-modal alignment score.** It is evident that CLIP@336 and CLIP@224 have the highest A scores. **Regarding the local representation accuracy reflected by the C score, DINOv2, as a self-supervised method, excels in local representation better than contrastive learning methods.** This is why DINOv2 achieves a relatively high score on the C score. So I have the following serious concerns:
> > * **In the experiment conducted by the authors, the optimal model combination is CLIP + DINOv2. However, the AC score naturally favors visual representations containing CLIP features, resulting in a high A score, while visual representations containing DINOv2 features score higher on the C score.** Unless AC score can be generalized to more combinations, such as SigLIP ViT-SO400M/14@384 + DINOv2 ViT-L/14@518 + OpenAI CLIP ViT-L/14@336 + ConvNeXt-XXL@1024, that won't dispel my concerns.
> > * The A and C scores for CLIP@224 are higher than those for CLIP@336, which, in my opinion, seems highly unreasonable. This further strengthens my concern that the AC score may be biased.
> >
> >
> > Finally, I suggest that the authors improve the layout of the manuscript. The current large gaps in the text may reduce the reader's interest.

---

> ### Author Response · Authors · 2024-12-03
>
> We wish to kindly emphasize that some of the statements of Reviewer 9bVh may reflect a misunderstanding of our work:
>
> **1. Comparison between SigLIP and CLIP -** It is never our intention to argue whether SigLIP or CLIP is a better model in terms of performance. In our work, we demonstrate that cross-alignment and correspondence of vision representations are positively correlated to the performance of MLLM. Therefore, **we consider a diverse range of vision representations that could be used for MLLMs, regardless of their sizes**. SigLIP and CLIP are merely data points that follow the law we discovered. This was added in Table 5 and 6 captions in the revised Appendix.
>
> **2. Generalization of AC Score - Our results show that the AC score generalizes to the combination of SigLIP ViT-SO400M/14@384 + DINOv2 ViT-L/14@518 + OpenAI CLIP ViT-L/14@336 + ConvNeXt-XXL@1024.** According to the configuration outlined by reviewer 9bVh, which is a recognized current SOTA work, the Cambrian's [1] setting, we have also referenced it in our paper. To address reviewer 9bVh's concerns more comprehensively, we take the official checkpoints from the authors of Cambrian and evaluated the AC score. Additionally, we associated the AC score with the evaluation on MMBench, MME, and MMMU. The results presented in Cambrian's paper (Table 3) is consistent with our AC score. As shown in the below table, the AC score successfully indicate the lowest and highest performing vision representations, with bold indicate the highest for each benchmark.
>
> | Model                     | A Score | C Score | MMBench       | MME           | MMM U          |
> |---------------------------|---------|---------|---------------|---------------|----------------|
> | CLIP@224                 | 81.5    | 14.3    | 64.18         | 1449.64       | 36.2           |
> |                           |         |         | AC score: 0.956 | AC score: 0.930 | AC score: 0.645 |
> | CLIP@336                 | 78.8    | 15.66   | 64.26         | **1502.70**       | 35             |
> |                           |         |         | AC score: 0.967 | **AC score: 0.990** | AC score: 0.727 |
> | C+D@224                  | 53.7    | 23.6    | **65.72**         | 1436          | **36.9**           |
> |                           |         |         | **AC score: 0.976** | AC score: 0.931 | **AC score: 0.725** |
> | C+D@336                  | 51.2    | 26.08   | 65.12         | 1475          | 34.6           |
> |                           |         |         | AC score: 0.951 | AC score: 0.836 | AC score: 0.577 |
> | SigLIP ViT-SO400M/14 + DINOv2 ViT-L/14@518 + OpenAI CLIP ViT-L/14@33 + ConvNeXt-XXL@102 (performance from Cambrian’s Table 3; AC score recalculated) | 49.902  | 21.07   | 63.32         | 1479.46       | 35.49          |
> |   |         |         | AC score: 0.924 | AC score: 0.884 | AC score: 0.721 |
>
> **3. Discussion on A and C Scores for CLIP@224 and CLIP@336 -**
>
> |                         | CLIP@224 | CLIP@336      |
> |-------------------------|----------|---------------|
> | Cross-modal Alignment   | 0.820    | 0.795 (-3.0%) |
> | Correspondence          | 14.3 (-8.7%) | 15.66      |
>
> The above table compares A and C scores between CLIP@224 and CLIP@336, showing that CLIP@224 has a significantly lower C score, while CLIP@336 experiences a slight drop in A score. Since CLIP has a high A score, and A scores are expected to exhibit diminishing returns as they approach the higher range, the slight drop is anticipated to have minimal effect on predicted performance.
>  - **CLIP@336’s higher C score than CLIP@224: This is reasonable as higher resolution better preserves visual details, leading to higher correspondence.**
>
> - **CLIP@224’s higher A score than CLIP@336: To the best of our knowledge, no prior work has measured the cross-modal alignment of the same contrastive learning model with different input resolutions.** Consequently, it is challenging to conclusively interpret the true cross-modality behavior in this case.
>
> We have added this discussion to revised Appendix 4.
>
> We wish to emphasize again that our work represents the first attempt to quantify cross-modality alignment in vision representations for MLLMs. Additionally, A score is intentionally designed to be flexible, and we encourage future works to refine it to eliminate potential biases. We believe our work has already made a significant contribution by proposing a meaningful correlation, which holds practical value for the future development of MLLMs.
>
> [1] Cambrian-1:A Fully Open, Vision-Centric Exploration of Multimodal LLMs.

---

### Official Review · Reviewer_Nwyo · 2024-11-03

**Soundness:** 3
**Presentation:** 3
**Contribution:** 3
**Rating:** 6
**Confidence:** 3

**Summary:**

This paper introduce a AC score that can be used to predict the performance of LVLM with specific vision encoders.

**Strengths:**

1. AC policy consistently predicts the optimal vision representation with minimal resources.
2. Provide a insight of the choice and design of vision encoder for LVLM.

**Weaknesses:**

1. The calculation of the A score by using the image feature of CLIP might not fully fulfill the motivation for quantifying the cross-modal alignment, as it assumes that the image-text features are perfectly aligned in CLIP.

**Questions:**

1. Please clarify the meaning and origin of the dimension 6 in  $X ∈ R^{k×6}$ in Policy Fitting. Is it related to the coefficients of the AC score?
2. Provide a pseudo code for the AC POLICY algorithm to enhance understanding.
3. Explain the choice of Recall@3 as the main metric and supplement with other metrics (like Recall@1, Recall@2, etc.).
4. Present the specific format (contains the weights and bias) of Policy Fitting and analyze the differences in fitting results across benchmarks.
5. SigLIP is supposed to have better cross-modal alignment as an improved version of CLIP, but it shows a lower score in your setting. Please explain.
6. Does a higher AC score always mean better performance?
7. Besides the A and C scores, are there other factors that need to be considered?

---

> ### Author Response · Authors · 2024-11-18
> **Reply to Reviewer Nwyo (1/3)**
>
> Thanks for your time in reviewing our work! Here, we address your concerns as following:
>
> ## Weaknesses:
>
> > *W1. The calculation of the A score by using the image feature of CLIP might not fully fulfill the motivation for quantifying the cross-modal alignment, as it assumes that the image-text features are perfectly aligned in CLIP.*
>
> While this limitation is acknowledged in the limitation section of our work, we want to note that the AC Policy is designed to establish only the relative order of vision representations within the search space. This means that we **only need to assume that CLIP has the highest cross-modal alignment compared to other vision encoders**, not "perfect alignment", for the policy to work. Moreover, the **reference model for the A score is designed to be flexible and can be adjusted** based on individual preferences, such as the belief that another model has the highest cross-modal alignment or to minimize fitting error.
>
> ## Questions:
>
> > *Q1. Please clarify the meaning and origin of the dimension 6 in $k^\prime$ in Policy Fitting. Is it related to the coefficients of the AC score?*
>
> Yes, the dimension 6 corresponds to the coefficients for the polynomial-transformed A and C scores. Without any transformation, fitting the A and C scores for each vision representation would involve two terms: $[A, C]$, in a linear regression. However, the AC score applies a second-degree polynomial transformation to the A and C scores, resulting in a six-term vector for fitting. In vectorized form, this is represented as $[1, A, C, A^2, A \cdot C, C^2]$.
>
>
> > *Q2. Provide a pseudo code for the AC POLICY algorithm to enhance understanding.*
>
> Thank you for the suggestion. We have included the calculations for the A and C scores, as well as the AC Policy, in Appendix Section 6.
>
> > *Q3. Explain the choice of Recall@3 as the main metric and supplement with other metrics (like Recall@1, Recall@2, etc.).*
>
> In Table 1 of the paper, we demonstrate that, on average, only $3.88$ finetuning runs are required to achieve $89.69$% Recall@3 in identifying the optimal vision representation. This means that, with a computation budget of at most $7$ runs, we can find the optimal vision representation in almost $90$% of the time. Using this same budget, we further evaluate other metrics, such as Recall@1 and Recall@2, to provide a comprehensive assessment.
>
> | Metric     | Average Finetuning Runs | Total Budget       | Chance of Finding Optimal VR |
> |------------|--------------------------|--------------------|------------------------------|
> | Random     | 6                        | 7 Runs (out of 13) | 48.2%                        |
> | Recall@1   | 6                        | 7 Runs             | 68.23%                       |
> | Recall@2   | 5                        | 7 Runs             | 81.01%                       |
> | Recall@3   | 3.88                     | 6.88 Runs          | 89.68%                       |
>
> We observe that, given the same computation budget where only 7 of vision representations in the search space can be tested, using Recall@1 provides a significant $20$% improvement over random trials in the chance of finding the optimal vision representation. However, using Recall@3 as a metric further optimizes this chance, demonstrating its practicality. This observation implies that, rather than assuming an absolute "optimal vision representation", we should acknowledge that each finetuning and inference process is subject to fluctuations, making "optimal" predictions inherently challenging. Employing Recall@3 accounts for these fluctuations, providing a more robust and practical approach without increasing computation costs.
>
> Additionally, we provide the average performance difference between the Top-1 and Top-2 vision representations trained MLLM to illustrate that, in most cases, the performance differences are minimal and their order can be attributed to fluctuations:
>
> | Benchmark    | Percentage Difference Between Top-1 and Top-2 VR |
> |--------------|---------------------------------------------|
> | MMBench      | 0.915%                                      |
> | MME          | 1.83%                                       |
> | OKVQA        | 1.39%                                       |
> | Seed-Bench   | 0.434%                                      |
> | MMMU         | 0.806%                                      |
> | TextVQA      | 0.269%                                      |
> | VizWiz       | 0.410%                                      |
> | ScienceQA    | 0.215%                                      |

---

> ### Author Response · Authors · 2024-11-18
> **Reply to Reviewer Nwyo (2/3)**
>
> > *Q4. Present the specific format (contains the weights and bias) of Policy Fitting and analyze the differences in fitting results across benchmarks.*
>
> | Benchmark    | Fitted Equation                                                                 |
> |--------------|---------------------------------------------------------------------------------|
> | MMBench      | $−0.127 + 3.191A + 0.910C − 1.721A^2 − 1.570A \cdot C − 0.189C^2$                        |
> | MME          | $−0.163 + 1.915A + 1.591C − 1.318A^2 − 0.073A \cdot C − 1.075C^2$                        |
> | OKVQA        | $−0.121 + 2.367A + 0.933C − 1.804A^2 + 0.156A \cdot C − 0.503C^2$                        |
> | SEED-Bench   | $0.938 + 1.740A + 1.345C − 1.271A^2 + 0.322A \cdot C − 0.856C^2$                         |
> | MMMU         | $−0.241 + 2.747A + 1.416C − 2.145A^2 − 0.242A \cdot C − 1.212C^2$                        |
> | TextVQA      | $−0.065 + 0.842A + 0.756C − 1.519A^2 + 2.943A \cdot C − 1.062C^2$                        |
> | VizWiz       | $0.069 + 2.145A − 0.102C − 1.772A^2 + 0.720A \cdot C + 0.116C^2$                         |
> | ScienceQA    | $0.013 − 0.198A + 1.422C − 0.487A^2 + 2.400A  \cdot C − 1.409C^2$                         |
>
> Common behavior observed across most benchmarks:
> - **Positive coefficient for $A$ and $C$ terms:** As A and C scores increase, benchmark performance increases proportionally to the coefficients of these terms.
> - **Negative coefficient for $A^2$ and $C^2$ terms:** This indicates a diminishing return or saturation effect—while increasing A and C initially improves performance, their positive impact decreases as the scores grow larger.
> - **Positive $A$ and $C$ interaction term ($A \cdot C$):** A positive coefficient for the interaction term implies that simultaneous increases in A and C magnify their combined effect on performance, resulting in a synergistic boost.
>
> Comparing across benchmarks, we observe that for vision-based benchmarks (MMBench, MME, OKVQA, SEED-Bench), the coefficients for A and C terms are consistently positive, indicating that cross-modal alignment and correspondence are positively correlated with performance. They all have a heavier weight on A than C. However, we also note that the effect diminishes as A and C scores continue to increase, eventually plateauing (at the maximum score for the benchmark). This is due to the negative coefficients for the square terms, which represent a saturation effect where performance gains slow down as A and C scores grow.
>
> For OCR-based benchmarks, such as VizWiz and ScienceQA, we observe unexpected coefficients. We attribute this to the issue discussed in Section 6.3. Specifically, our correspondence calculation image set is based on natural images rather than images containing text, which leads to inaccuracies in measuring correspondence for these tasks. As a result, the fitted function is less effective for OCR-based benchmarks.
>
> > *Q5. SigLIP is supposed to have better cross-modal alignment as an improved version of CLIP, but it shows a lower score in your setting. Please explain.*
>
> SigLIP is a widely adopted vision encoder for MLLMs and has been shown to outperform CLIP on certain benchmarks under specific training data settings. However, this improved performance is not guaranteed to generalize across all settings. In our specific setting, as shown in Appendix 3, SigLIP demonstrates lower performance, which aligns with its lower A score measurement.
>
> More importantly, to the best of our knowledge, no existing work has directly measured the cross-modal alignment between CLIP and SigLIP, and the original SigLIP paper does not mention any explicit design improvements for cross-modal alignment. As Reviewer 9bVh believes that BLIP is a stronger contrastive learning model than CLIP, this shows that cross-modal alignment remains an active area of research, and researchers have different beliefs on cross-modal alignment of different vision representations. Notably, our A score calculation framework is the first attempt in the MLLM field to quantify this factor.
>
> Additionally, we conduct experiment using SigLIP as the reference embedding for the A score. If SigLIP is supposed to have better cross-modal alignment than CLIP, it would result in more accurate A scores and yield similar or higher fitting correlation coefficients. However, we observed a $3.73$% drop in the fitting $R^2$, indicating that SigLIP may be less reliable for measuring cross-modal alignment, at least under the current framework for quantifying it. This result suggests that SigLIP’s cross-modal alignment is questionable in this context. More detail, please see [our response to Reviewer 9bVh’s Question 1](https://openreview.net/forum?id=SZm3hxmksx&noteId=cTux9cXqGF).

---

> ### Author Response · Authors · 2024-11-19
> **Reply to Reviewer Nwyo (3/3)**
>
> > *Q6. Does a higher AC score always mean better performance?*
>
> Yes, the AC score and performance exhibit a linear relationship, meaning that as AC score increases, performance improves proportionally. Furthermore, we test various alternative relationships between the AC score and performance, confirming that the current conclusion is indeed the most accurate. For more details, please refer to our response to [Reviewer JSYS’s Weakness 2](https://openreview.net/forum?id=SZm3hxmksx&noteId=68I1ntdoqk).
>
> > *Q7. Besides the A and C scores, are there other factors that need to be considered?*
>
> So far, we have no evidence suggesting that other factors need to be considered. For instance, we tested the effect of changing the LLM and found that the correlation between AC scores and model performance still holds. This indicates that the choice of LLM is not a significant factor. Please see [our response to Reviewer JSYS’s Weakness 1](https://openreview.net/forum?id=SZm3hxmksx&noteId=68I1ntdoqk) for more details.
>
> Additionally, we investigate the relationship between vision encoder parameters and performance in [our response to Reviewer 9bVh’s Weakness 3](https://openreview.net/forum?id=SZm3hxmksx&noteId=cTux9cXqGF) and found no correlation. However, we encourage the community to explore additional factors that could build upon our findings, whether specific to certain benchmarks or applicable more broadly.

---

> ### Author Response · Authors · 2024-11-20
> **Follow-Up on Rebuttal for Your Feedback**
>
> Dear Reviewer Nwyo,
>
> Thank you once again for the time and effort you’ve dedicated to reviewing our work. We have carefully replied to your seven questions you highlighted in your feedback and provided additional experiments as needed.
>
> We kindly ask if you could let us know whether our responses have sufficiently addressed your concerns. Your continued feedback would be invaluable in helping us further refine the paper during the discussion phase.

---

### Official Review · Reviewer_AFzz · 2024-11-03

**Soundness:** 3
**Presentation:** 2
**Contribution:** 2
**Rating:** 6
**Confidence:** 3

**Summary:**

In this paper, the authors analyze the relationship between the cross alignment and correspondence (AC) score and MLLM performance. The AC score is a second-degree polynomial transformation of the alignment and correspondence score. Through a few training runs, the AC Policy can be determined and subsequently used to predict the optimal configurations of MLLMs. Thus, the AC Policy can reduce computational costs when deciding the optimal vision representations for MLLMs.

**Strengths:**

1. The experiments are well-conducted and quite comprehensive.
2. The analyses of A, C, and AC scores, as well as their correlation with model performances are quite clear and valuable, highlighting the effectiveness of the proposed AC score.

**Weaknesses:**

1. How should we choose the number of $k'$ for Policy fitting? In Table 1, each benchmark has its own number of finetuning runs required to successfully predict the optimal vision representation. While 3 is the most common number across the benchmarks, there are still some outliers. It is difficult to determine the exact number of $k'$ for a benchmark without having the groundtruth. This suggests that the AC policy may not be very practical.
2. All the experiments are conducted under the condition of freezing the Vision Encoders. The conclusions may change when the Vision Encoders are unfrozen, which limits the generalizability of the AC Policy and these conclusions.
3. It is confusing that “if the vision representation ensures accurate correspondence, more visual details are considered even if not attended by the text token”. This statement implies that a high correspondence vision representation preserves visual details, which is ungrounded.

**Questions:**

1. In section 3.2, the explanation of key points and their prediction process is missing.
2. In Table 1, could you provide the top-1/2 predictions for a more comprehensive analysis?
3. Why does the 13 setting only require 12 finetuning runs to predict the optimal vision representations in a random setting?

---

> ### Author Response · Authors · 2024-11-18
> **Reply to Reviewer AFzz (1/3)**
>
> Thanks for your time in reviewing our work! Here, we address your concerns as following:
>
> ## Weaknesses:
>
> > *W1-1. How should we choose the number of $k^\prime$ for Policy fitting?*
>
> We should choose the number of $k^\prime$ based on the available computation budget. If one has the time and budget to test $10$ vision representations, they should test $10$; if they are willing to finetune for $100$ runs, they should do $100$. Ideally, we would hope to test every single vision representation in existence, which would guarantee finding the optimal vision representation for MLLM. However, the vast number of representations and the complexity of feature combinations make the search space incredibly large. In the real world, everyone tests only a subset of vision representations for MLLM, sacrificing chances of finding the optimal representation. This is why **our work stands on a practical assumption: one should finetune the number of MLLMs based on their maximum budget, and AC Policy is designed to maximize the chance of finding an optimal vision representation for their MLLM.**
>
> > *W1-2. Given the outliers in the number of finetuning across benchmarks, on what kind of benchmark should we allocate the more computation budget for AC Policy?*
>
> Based on our answer to question 1, we would like to address your concern about the outliers in the number of $k^\prime$ by approaching it in the context of this question. I suggest allocating more finetuning runs if the task evaluation set differs significantly from the correspondence score calculation set. For example, if the benchmark is testing the OCR ability of an MLLM while the correspondence dataset evaluates correspondence on natural images, the performance correlation would be lower. As discussed in Section 6.3, benchmark performances are less correlated in OCR-based tasks because the correspondence dataset measures the correspondence of natural images. Consequently, the prediction of the optimal vision representation would be less accurate, thus requiring more runs to find the optimal representation. To make the AC Policy more practical, we not only provided an explanation for the outlier but also suggest that any future contributions include correspondence evaluation sets specifically for images containing text.
>
> > *W2. All the experiments are conducted under the condition of freezing the Vision Encoders. The conclusions may change when the Vision Encoders are unfrozen, which limits the generalizability of the AC Policy and these conclusions.*
>
> To ensure scientific rigor in framing the Law of Vision Representation in MLLM, we clearly state the assumptions and conditions under which it holds. **We emphasize that unfrozen vision encoders fall outside the scope of the Law of VR, as this scenario breaks both our assumptions and the framing of the question** in two key ways:
>
> 1. **Theoretical Aspect:** We assume that the vision representation is the only independent variable, while the alignment module and LLM architecture remain fixed (line 145). However, with an unfrozen vision encoder, we cannot guarantee that the vision encoder does not take the function of the alignment module. This causes the architecture and role of the alignment module to change alongside the encoder, making the experiment uncontrolled and the models no longer comparable.
>
> 2. **Application Aspect:** A key application of the law is to determine the optimal vision representation before training the MLLM. With an unfrozen vision encoder, the vision representation evolves during stage 2 [1] or even up to the final stage [2, 3]. This makes it impossible to identify the optimal vision representation without completing the full training process. While it would be interesting to explore the scenario of unfrozen encoders, this case and exploration lacks practical use because people need to train until the final stage anyways.
>
> Therefore, our conclusions are framed specifically within the context of frozen vision encoders, and the AC Policy is designed to operate under these assumptions.
>
> [1] Qwen-VL: A Versatile Vision-Language Model for Understanding, Localization, Text Reading, and Beyond.
>
> [2] LLaVA-OneVision: Easy Visual Task Transfer.
>
> [3] Cambrian-1: A Fully Open, Vision-Centric Exploration of Multimodal LLMs.

---

> ### Author Response · Authors · 2024-11-18
> **Reply to Reviewer AFzz (2/3)**
>
> > *W3. It is confusing that “if the vision representation ensures accurate correspondence, more visual details are considered even if not attended by the text token”. This statement implies that a high correspondence vision representation preserves visual details, which is ungrounded.*
>
> **We would like to point out that it is grounded that accurate correspondences in vision representation mean that more visual details are considered.** Finding correspondences between images has been a core topic in computer vision for decades. It focuses on identifying semantically and geometrically similar points across different images given any keypoints. This goal aligns closely with the purpose of learning visual representations that many researchers in computer vision aim to achieve. Back to classic methods like SIFT [1] to modern powerful visual encoders like DINOV2 [2], researchers have worked to create representations that are robust to changes in appearance, visual illumination, local deformation, and viewpoints, all of which aligns the goal of finding correspondences. To this end, checking correspondences is now a key method to evaluate the ability of understanding visual cues in the representations, as seen in works like [3, 4]. It is also used as an objective for representation learning, as demonstrated by methods like [4, 5].
>
> [1] Distinctive Image Features from Scale-Invariant Keypoints. IJCV 2004
>
> [2] DINOV2: Learning Robust Visual Features without Supervision.
>
> [3] Probing the 3D Awareness of Visual Foundation Models. CVPR 2024
>
> [4] 3DiffTection: 3D Object Detection with Geometry-Aware Diffusion Features. CVPR 2024
>
> [4] Unsupervised Learning of Visual Representations by Solving Jigsaw Puzzles. ECCV 2016
>
> [5] SuperPoint: Self-Supervised Interest Point Detection and Description. CVPR 2018
>
> ## Questions:
>
> > *Q1. In section 3.2, the explanation of key points and their prediction process is missing.*
>
> We have added key points prediction process as part of the C score calculation, included as pseudocode Appendix Section 6. The key points in correspondence finding refer to distinctive points or regions in an image that can be identified and matched across different images. For visualizations of key points, refer to [[1]’s website](https://telling-left-from-right.github.io/). In our work, we directly use the key points provided by the SPair-71K dataset and compute the correspondences based on them.
>
> [1] Telling Left from Right: Identifying Geometry-Aware Semantic Correspondence. CVPR 2024

---

> ### Author Response · Authors · 2024-11-18
> **Reply to Reviewer AFzz (3/3)**
>
> > *Q2. In Table 1, could you provide the top-1/2 predictions for a more comprehensive analysis?*
>
> In the paper Table 1, we demonstrate that, on average, only $3.88$ finetuning runs are required to achieve $89.69$% Recall@3 in identifying the optimal vision representation. This means that, with a computation budget of at most $7$ runs, we can find the optimal vision representation almost $90$% of the time. Using this same budget, we further evaluate other metrics, such as Recall@1 and Recall@2, to provide a comprehensive assessment.
>
> | Metric     | Average Finetuning Runs | Total Budget       | Chance of Finding Optimal VR |
> |------------|--------------------------|--------------------|------------------------------|
> | Random     | 6                        | 7 Runs (out of 13) | 48.2%                        |
> | Recall@1   | 6                        | 7 Runs             | 68.23%                       |
> | Recall@2   | 5                        | 7 Runs             | 81.01%                       |
> | Recall@3   | 3.88                     | 6.88 Runs          | 89.68%                       |
>
> We observe that, given the same computation budget where only $7$ of vision representations in the search space can be tested, using Recall@1 provides a significant $20$% improvement over random trials in the chance of finding the optimal vision representation. However, using Recall@3 as a metric further optimizes this chance, demonstrating its practicality. This observation implies that, rather than assuming an absolute "optimal vision representation", we should acknowledge that each finetuning and inference process is subject to fluctuations, making "optimal'" predictions inherently challenging. Employing Recall@3 accounts for these fluctuations, providing a more robust and practical approach without increasing computation costs.
>
> Additionally, we provide the average performance difference between the Top-1 and Top-2 vision representations trained MLLM to illustrate that, in most cases, the performance differences are minimal and their order can be attributed to fluctuations:
> | Benchmark    | Percentage Difference Between Top-1 and Top-2 VR |
> |--------------|---------------------------------------------|
> | MMBench      | 0.915%                                      |
> | MME          | 1.83%                                       |
> | OKVQA        | 1.39%                                       |
> | Seed-Bench   | 0.434%                                      |
> | MMMU         | 0.806%                                      |
> | TextVQA      | 0.269%                                      |
> | VizWiz       | 0.410%                                      |
> | ScienceQA    | 0.215%                                      |
>
> > *Q3. Why does the 13 setting only require 12 finetuning runs to predict the optimal vision representations in a random setting?*
>
> First of all, in a random setting, we need to run all to find the optimal vision representation, but running all 13 out of 13 settings is no longer a prediction task. To maintain a fair comparison under the prediction problem framework, we use a probabilistic baseline. Specifically, we set the acceptable prediction Recall@3 at approximately $90$%. To achieve this level of performance in the random setting, 12 finetuning runs are required.

---

> ### Author Response · Authors · 2024-11-20
> **Follow-Up on Rebuttal for Your Feedback**
>
> Dear Reviewer AFzz,
>
> Thank you once again for the time and effort you’ve dedicated to reviewing our work. We have carefully replied to the three weaknesses and three questions you highlighted in your feedback and provided additional experiments as needed.
>
> We kindly ask if you could let us know whether our responses have sufficiently addressed your concerns. Your continued feedback would be invaluable in helping us further refine the paper during the discussion phase.

---

> > ### Comment · Reviewer_AFzz · 2024-11-20
> >
> > Thanks for the detailed response. The reviewers addressed some raised points and added additional results supporting their conclusions. Based on their feedback I will increase my score.
> >
> > However, I strongly suggest that the authors add more detailed explanations of the underlying assumptions (e.g., the unfrozen ViT and choosing K' based on the maximum budget), as well as analyses of the limitations when the task evaluation set significantly differs from the correspondence score calculation set in a revised version.

---

> > > ### Author Response · Authors · 2024-11-20
> > >
> > > Thank you for your feedback on our rebuttal. We greatly appreciate your suggestions. We are working on the revised version and will ensure that the revised version addresses your points in detail on unfrozen ViT, choosing k', and analysis of limitation on C score.

---

### Official Review · Reviewer_JSYS · 2024-11-07

**Soundness:** 2
**Presentation:** 2
**Contribution:** 2
**Rating:** 5
**Confidence:** 4

**Summary:**

The paper presents the Law of Vision Representation in multimodal large language models (MLLMs), aiming to establish a strong relationship between vision representation quality, cross-modal alignment, and model performance. The authors introduce the AC score to quantify cross-modal Alignment and Correspondence, demonstrating a linear correlation between AC scores and MLLM performance across multiple benchmarks. The AC policy is also proposed as a method to efficiently select optimal vision representations, potentially reducing computational costs and energy consumption significantly compared to traditional methods.

**Strengths:**

1. The paper identifies the limitations in current methods for choosing vision representations, highlighting the need for a deeper understanding rather than an empirical, trial-based approach.
2. The proposal of the Law of Vision Representation is an innovative concept that attempts to quantify the relationship between cross-modal alignment, correspondence in vision representation, and MLLM performance through the AC score.
3. The paper highlights an approach that achieves computational efficiency by reducing the need for finetuning with each vision representation change. This leads to a substantial reduction in computational cost.

**Weaknesses:**

While this work raises an important question in MLLM, there are certain weaknesses that prevent me from assigning a higher score.
1. The experiments in this paper are conducted on a specific MLLM setup, with a particular LLM and projector configuration, which raises doubts about whether the conclusions can be applied if the LLM or projector settings are changed. Given that the paper aims to establish a "law of vision representation," the current experiments seem too limited in scope to substantiate such a universal law.  Additional experiments should be carried out using various LLM sizes and projector designs to showcase wider applicability.
2. The choice of linear regression for modeling the relationship between the AC score and MLLM performance lacks sufficient explanation. It is unclear why a linear regression model is specifically used, and there is no discussion on whether other types of regression (e.g., polynomial, logarithmic) could potentially better capture this relationship. A justification for the linear assumption or a comparison with other regression models would strengthen the analysis. A comparison of various regression models (such as linear, polynomial, and logarithmic) should be performed, and their respective performance metrics should be reported to support the selection of linear regression.
3. Although the authors provide a theoretical explanation of the impact of the AC score on performance, the paper would be strengthened by a more rigorous proof or formal derivation of the mathematical relationship. Additionally, a more in-depth discussion on the design and application of AC scores, encompassing both explainable and theoretical aspects, should be included.

**Questions:**

1. Typo in Table 1 'Number of Fintuning' -> 'Number of Finetuning'.
2. The experimental setting for the chosen LLM and projector is unclear. Could you give more information about the experimental setting?

---

> ### Author Response · Authors · 2024-11-18
> **Reply to Reviewer JSYS (1/2)**
>
> Thanks for your time in reviewing our work! Here, we address your concerns as following:
>
> ## Weakness:
>
> > *W1. Additional experiments should be carried out using various LLM sizes and projector designs to showcase wider applicability.*
>
> In this paper, we demonstrate the relationship between the AC score and performance by fitting a regression model and reporting the $R^2$ value. As shown in Table 3 of the paper, the $R^2$ achieves a strong $95.72%$ when averaged across four vision benchmarks under the LLaVA1.5 setting, where the LLM is Vicuna-7B-1.5.
>
> | LLM               | $R^2$ (Vision) | $R^2$ (OCR) |
> |--------------------|---------------------|------------------|
> | Vicuna-7B-1.5     | 95.72%              | 85.21%           |
> | Llama2-7B         | 98.01%              | 87.91%           |
> | Vicuna-13B-1.5    | 95.17%              | 88.50%           |
>
> In the above additional experiments, we show that the fitting $R^2$ remains strong, and in some cases even higher, when using different LLM types and sizes. The variation in $R^2$ falls within a reasonable range, indicating that the effect of LLM and vision representation compatibility, if it exists, is negligible compared to the influence of the A and C factors. These results demonstrate that **the Law of Vision Representation relationship holds across different LLMs**.
>
> We strongly believe that the Law of Vision Representation applies to a broader range of LLMs and projectors. This is because the vision encoder serves as the initial feature extractor for vision information, and changes in the alignment module or LLM do not alter the quality of the information extracted. While the empirical validation of the Law could extend infinitely, finetuning and inference MLLMs is expensive, with the settings tested in this paper and rebuttal costing nearly $100,000. This underscores the significance of the Law of Vision Representation, as it enables others to avoid these high costs. We hope that the additional experiments provided convey the broad applicability of our findings.
>
> > *W2. A justification for the linear assumption or a comparison with other regression models would strengthen the analysis.*
>
> In our work, we use a linear regression model to capture the relationship between the AC score and model performance. To clarify, the AC score is a second-degree polynomial transformation of the A and C scores. This means that we are effectively applying polynomial regression through the transformation of A and C before fitting a linear regression model to predict performance.
>
> As shown in the paper Table 3, we compare the results of using no transformation (simple linear regression) and polynomial transformation (polynomial transformation + linear regression = polynomial regression), demonstrating that the latter provides better predictive performance. To strengthen our analysis, we conducted additional experiments using other regression models. Here, we calculated the train and validation mean square error (MSE) using a leave-one-out cross-validation (LOOCV) strategy on 13 vision representations, averaged across 4 vision-based benchmarks. These results further validate the effectiveness of our approach.
>
> | Regression Model                                      | Train MSE | Validation MSE |
> |-------------------------------------------------------|-----------|----------------|
> | Linear (corresponds to Table 3 no transformation)     | 0.0229    | 0.0358         |
> | Degree 2 Polynomial (corresponds to ours; Table 3 polynomial transformation) | 0.0056    | 0.0215         |
> | Degree 3 Polynomial                                   | 0.0017    | 0.0989         |
> | Logarithmic                                           | 0.0177    | 0.0290         |
>
> By observing and trading off between train and validation MSE, we find that our method best captures the relationship between the AC score and model performance. Simple linear regression without polynomial transformation shows underfitting as shown in high train MSE. In contrast, higher-degree polynomial regression on A and C leads to overfitting as shown in high validation MSE. Logarithmic regression, on the other hand, exhibits high train MSE, indicating that it does not accurately model the relationship. These findings support our choice of using a second-degree polynomial transformation combined with linear regression as the most appropriate approach.
>
> Additionally, the use of linear regression on top of the polynomial transformation of the A and C scores offers excellent explainability by allowing direct inspection of the weights and biases. For more details, please refer to [our response to Reviewer Nwyo’s Question 4](https://openreview.net/forum?id=SZm3hxmksx&noteId=9ntxX3H0aN). **In summary, we selected our current modeling technique based on its strong empirical performance and superior explainability.**

---

> ### Author Response · Authors · 2024-11-18
> **Reply to Reviewer JSYS (2/2)**
>
> > *W3. Although the authors provide a theoretical explanation of the impact of the AC score on performance, the paper would be strengthened by a more rigorous proof or formal derivation of the mathematical relationship. Additionally, a more in-depth discussion on the design and application of AC scores, encompassing both explainable and theoretical aspects, should be included.*
>
> We want to emphasize that ours is the first work in the MLLM field to attribute model performance to specific network design characteristics. Empirically, we demonstrated that the AC score is linearly correlated with model performance, achieving a strong correlation with an $R^2$ of $95.72$. In addition, we provide a theoretical explanation that the AC score is positively correlated with MLLM performance. Influential works such as the Scaling Laws [1, 2] also adopt an empirical approach, demonstrating relationships without formal proofs. Therefore, **while we recognize the value of a formal derivation, we are confident that our findings provide valuable insights, even in the absence of a formal mathematical derivation.**
>
> [1] Scaling Laws for Neural Language Models.
>
> [2] Training Compute-Optimal Large Language Models.
>
> ## Questions:
>
> > *Q2. The experimental setting for the chosen LLM and projector is unclear. Could you give more information about the experimental setting?*
>
> We follow the experimental settings used in LLaVA 1.5, which is one of the most widely adopted and generalizable MLLM architectures. Specifically, the base LLM is Vicuna-7B, and the projector consists of two linear layers with a GeLU activation in between.

---

> ### Author Response · Authors · 2024-11-20
> **Follow-Up on Rebuttal for Your Feedback**
>
> Dear Reviewer JSYS,
>
> Thank you once again for the time and effort you’ve dedicated to reviewing our work. We have carefully replied to the three weaknesses you highlighted in your feedback and provided additional experiments as needed.
>
> We kindly ask if you could let us know whether our responses have sufficiently addressed your concerns. Your continued feedback would be invaluable in helping us further refine the paper during the discussion phase.

---

> > ### Comment · Reviewer_JSYS · 2024-11-22
> >
> > Thank you for the detailed response. My primary concern is that the paper claims to have discovered a representation law for MLLMs. However, all the experiments, including the additional ones, are conducted solely on the typical LLaVA1.5 architecture. The field of MLLMs is much broader, encompassing diverse types of projectors and architectures, such as LLaVA-OneVision and Flamingo. While the discovery of the AC score is innovative, I think the authors have overstated its contributions and significance. My comment is based on the perspective of whether this work truly establishes a universal visual representation law for MLLMs. From this standpoint, I see it as a very preliminary exploration.
> > Additionally, I kindly suggest that the authors revise the structure and writing of the paper. The current presentation, particularly regarding the experimental setup and conclusions, lacks clarity, which made it challenging for me to follow the content during my review.

---

> > > ### Author Response · Authors · 2024-12-03
> > >
> > > We appreciate reviewer JSYS finds the discovery of the AC score innovative, and we acknowledge that everyone has a different assessment of our contribution. However, we wish to make a small correction to reviewer JSYS’s response:
> > >
> > > Our experiments are not necessarily conducted solely on the typical LLaVA 1.5 architecture: LLaVA 1.5 architecture is well adopted and renamed into a large category. Therefore, our findings apply to a popular category instead of a single architecture. Following NVLM [1], we categorize MLLMs into the following types: (1) Decoder-only MLLMs: These MLLMs consist of vision encoder(s) and an alignment module, such as a multilayer perceptron (MLP), which maps the vision representation into vision tokens. These tokens are designed to have a similar distribution as language tokens and are directly input into a language model in the same manner as language tokens. (2) Cross-attention-based MLLMs: These MLLMs include vision encoder(s) and an additional module, often serving as a downsampling component, such as a perceiver resampler. The vision tokens generated are integrated into the language model through cross-attention mechanisms.
> > >
> > > The Law of Vision Representation specifically focuses on decoder-only MLLM architecture due to their widespread adoption and their simplicity, which facilitates controlling variables in training recipes and enables clear mathematical modeling. Your mentioned LLaVA-OneVision is the same architecture as LLaVA 1.5, as they all lie in the category of decoder-only MLLMs. Flamingo lies in the category of cross-attention-based MLLMs. We have added this point as the assumption in Section 3.1.
> > >
> > > Additionally, we have revised experiential setup at Section 5.1 and added conclusion at Section 8.
> > >
> > > [1] NVLM: Open Frontier-Class Multimodal LLMs

---

### Author Response · Authors · 2024-11-18
**Reply to All Reviewers**

We carefully read and sincerely appreciate the reviewers’ comprehensive suggestions. We are really grateful to see the reviewers find our work as an **innovative concept** (JSYS) and **well-motivated** (9bVh), almost all of the reviewers (JSYS, Nwyo, 9bVh) agree with the **contribution of computation reduction for finetuning MLLM**. Specifically, reviewer (JSYS) **agree with the need of exploring and deepen understanding of current methods for choosing vision representation**. We also appreciate reviewer (AFzz) appreciate our **extensive experiments**.

In this work, we reveals a strong correlation between the combination of cross-modal alignment, correspondence in vision representation, and MLLM performance, we quantify these factors as cross-modal Alignment and Correspondence score (AC score), and through extensive experiments, we find the relationship between AC score and model performance.

We encourage active and multi-round discussions, so we will have the final updated PDF at the end of discussion phase, which will be merging all the reviewers’ suggestions. In this round, we added pseudo code of A, C score calculation, as well as AC Policy in Appendix Section 6 in blue text.

**Several common misunderstandings that we want to highlighted here:**

> *1. How does the law behave on settings other than LLaVA 1.5, such as different LLMs? (Reviewer JSYS, 9bVh)*

By adding extra experiments on additional type and size of LLM, we show that the fitting $R^2$ remains strong, and in some cases even higher, when using different LLM types and sizes. The variation in $R^2$ falls within a reasonable range, indicating that the effect of LLM and vision representation compatibility, if it exists, is negligible compared to the influence of the A and C factors. These results demonstrate that the Law of Vision Representation relationship holds across different LLMs.

> *2. Concern on CLIP embedding as reference embedding in A score. (Reviewer Nwyo, 9bVh)*

In our paper, we aim to elucidate the relationship between performance, alignment, and correspondence. We employ "A score" as an initial approach to quantify the concept of alignment in the field of MLLM. A score is designed to be flexible and can be adjusted based on individual preferences, such as the belief that another model has the highest cross-modal alignment or to minimize fitting error.

Addressing the concerns raised by Reviewers Nwyo and 9bVh, Nwyo think SigLIP is the better model, and 9bVh think BLIP should be a good try. We acknowledge that different researchers may have varying preferences for the strongest contrastive vision encoder. Nevertheless, many papers published post-CLIP have compared their results with CLIP across numerous benchmarks, thereby demonstrating the widespread acceptance of CLIP.

It's worth noting that most vision representations are not specifically designed to optimize the concept of alignment, hence there's no consensus on which model performs the best in this regard. Part of our proposal of A score seeks to establish a baseline method for quantifying cross-modal alignment, making CLIP an apt choice widely accepted by researchers.

We would like to compare additional settings, but as mentioned in our paper, the experiments are costly and time-consuming. We have included the SigLIP experiments in our response, which demonstrate that SigLIP has a poorer fitting result compared to CLIP. Therefore, we assert that selecting CLIP for our settings and experiments remains the optimal choice.

> *3. Why not evaluate AC Policy Recall@1 and Recall@2. Why Recall@3? (Reviewer AFzz, Nwyo)*

There maybe confusions on why Recall@3 is used. As we showed in the response, we provide the MLLM performance percentage difference between the Top-1 and Top-2 vision representations to illustrate that, in most cases, the performance differences are minimal and their order can be attributed to fluctuations. We also added the Recall@1 and Recall@2 in the response. We observe that, given the same computation budget where only 7 MLLM finetunings, using Recall@1 provides a significant 20% improvement over random trials in the chance of finding the optimal vision representation. However, using Recall@3 as a metric further optimizes this chance, demonstrating its practicality. This observation implies that, rather than assuming an absolute "optimal vision representation", we should acknowledge that each finetuning and inference process is subject to fluctuations, making "optimal" predictions inherently challenging. Employing Recall@3 accounts for these fluctuations, providing a more robust and practical approach without increasing computation costs.

---

### Meta-Review · Area_Chair_QUQS · 2024-12-20

**Metareview:**

Paper was reviewed by four expert reviewers and received 2 x marginally below the acceptance threshold and 2 x marginally above the acceptance threshold ratings. While generally the reviewers liked the premise of the work, they also pointed out a number of concerns, including (1) over-claiming of the results, as they only focus on a particular class of LLaVA models [JSYS]; (2) lack of clear exposition (pointed out by multiple reviewers, e.g., with respect to experimental setup and conclusions [JSYS] and underlying assumptions [AFzz]); and (3) lack of justification for the design beyond empirical evidence and verification that such design is adequate for border class of models, e.g., BLIP that has more complex transformer-based alignment and losses was not explored. Authors have provided comprehensive rebuttal addressing these issues and reviewers found some of the responses satisfactory. However, during the discussion, it was apparent that some concerns remained (e.g., concerned of limited scope that may prevent generalization of results to a broader range of MLLMs).

AC has read the reviews, rebuttal and the discussion, as well as the paper itself. AC agrees that the overall idea of the work is intriguing, i.e., to be able to select visual representation / encoder that is best suited for an MLLM. However, AC also agrees that (i) paper, even after revisions, is difficult to read and understand (despite the underlying concept being relatively simple), and (ii) further investigation is need to identify the full extent of the findings and whether what is presented could really be called a "Law". Speaking from the formal definition, a "law" is a statement that describes or predicts a range of phenomena. Certainly the findings in the paper suggest that AC score predicts a phenomena, but fall short of the "range" implied. Authors are encouraged to revise the manuscript to improve quality and to conduct additional experiments with architectures that differ by form of projection and losses for resubmission to the next top-tier venue.

**Additional Comments On Reviewer Discussion:**

Short discussion has followed the rebuttal. Overall, it appears that reviewers are still not fully convinced by the work and, while rebuttal has lead to some improvement of scores, no reviewer was willing to champion the paper. This lack of conviction by the reviewers has played into the final decision as noted above.

---

### Decision · Program_Chairs · 2025-01-22

Reject